# Estimating Unbiased Averages of Sensitive Attributes without Handshakes among Agents

## Abstract

We consider the problem of distributed averaging of sensitive attributes in a network of agents without central coordinators, where the graph of the network has an arbitrary degree sequence (degrees refer to numbers of neighbors of vertices). Usually, existing works solve this problem by assuming that either (i) the agents reveal their degrees to their neighbors or (ii) every two neighboring agents can perform handshakes (requests that rely on replies) in every exchange of information. However, the degrees suggest the profiles of the agents and the handshakes are impractical upon inactive agents. We propose an approach which solves the problem with privatized degrees and without handshakes upon a stronger self-organization. In particular, we propose a simple gossip algorithm that computes averages that are biased by the variance of the degrees and a mechanism that corrects that bias. We will suggest a use case of the proposed approach that allows for fitting a linear regression model in a distributed manner, while privatizing the target values, the features and the degrees. We will provide theoretical guarantees that the mean squared error between an estimated regression parameter and a true regression parameter is $\mathcal{O}(\frac{1}{n})$, where $n$ is the number of agents. We will show on synthetic graph datasets that the theoretical error is close to its empirical counterpart. Also, we will show on synthetic graph datasets and real graph datasets that the regression model fitted by our approach is close to the solution when locally privatized values are averaged by central coordinators.

## 1 Introduction

Over the last decade there has been significant interest in self-organizing distributed systems, where the nodes (agents) in a communication network collaborate without central coordinators. One basic task corresponds to the problem where every agent has an individual value and every agent would like to know the average of those values. As we will argue, existing works usually assume a form of handshakes between every two neighboring agents in every exchange of information, i.e., when one agent uses information of a second agent, this second agent becomes aware of this and must actively help the process. However, there exist practical scenarios where such interaction is time consuming due to inactivity of some agents.

To give an illustrative example on communication without handshakes, let us consider a group of researchers who aim at solving a particular problem. The researchers can follow each other, and

thus read each other's currently best strategy in each other's most recently published paper. The researchers work on their solution strategy individually and without necessarily directly contacting others, attempting to improve their current solution strategies based on their individual skills and the ideas read in the papers of the followed colleagues. They hope that at some point one of them will find a fully satisfactory solution.

Regarding our setting, we model the agents by a graph where neighboring agents are connected by edges. The agents have sensitive attributes, such as their degree (the number of neighbors), which they aim to privatize before sharing them with other agents as otherwise the sensitive attributes would suggest their profile. We remark that the agents hide (add appropriate noise to) their sensitive attributes on their own (without central coordinators).

Regarding our approach, the agents commit to a simple gossip algorithm for distributed averaging. In particular, every agent hides their sensitive attributes under differential privacy noise (differential privacy is a conventional guarantee of data privacy) and makes them visible to the neighbors. Then, every agent repeatedly computes the averages over the values revealed by their neighbors and updates the displayed value by the computed average. At some point this will converge, and the computed averages will result in the averages that are biased by the variance in the degrees (the averages are biased when the graph is non-regular). We will provide a mechanism that combines such biased averages so that the bias is corrected. We remark that our approach is characterized by self-organization: handshake-free interaction and asynchrony.

We will suggest a use case of our approach for fitting a simple linear regression model, while privatizing the target values, the features and the degrees. We will provide theoretical guarantees that the mean squared error between an estimated regression parameter and a true regression parameter is $\mathcal{O}(\frac{1}{n})$, where $n$ is the number of agents. We will show on synthetic graph datasets that the theoretical error is close to its empirical counterpart. Also, we will show on synthetic graph datasets and real graph datasets that the regression model fitted by the unbiased averages computed by our strategy is close to the solution that relies on central coordinators.

We briefly motivate several elements of our setting. Usually, there are two common types of information flow (Giakkoupis, 2011): pulling, where an agent asks its neighboring agent for its value, and pushing, where an agent sends its value to a neighboring agent. In this paper, we study a weak form of pulling where an agent obtains the current value of a neighbor without the neighbor being aware of this. In particular, every agent continuously publishes its current values so that its neighbors can obtain it, but there is no other communication (e.g., there is no communication process to build overlay networks (Jelasity et al., 2009) that can improve distributed computations). Since the agents do not exchange handshakes (like in the Transport Layer Security protocol), information dissemination is more robust against inactive agents. Further, we assume that the degree is a sensitive attribute because in some contexts it suggests the profile of an agent (Hay et al., 2009; 2010). Finally, we mention that our approach applies for graphs with power-law degree sequences which, as suggested by Zipf's law, are common in real-world (e.g., computer, social, biological) networks.

**Outline.** In Section 2, we precise our setting. In Section 3, we provide the literature study. In Section 4, we state our approach. In Section 5, we relate our approach to a use case on linear regression. In Section 6, we provide theoretical guarantees for the mean squared error between an estimated regression parameter and a true regression parameter. In Section 7, we discuss the experiments on synthetic graph datasets and real graph datasets. In Section 8, we conclude.

## 2 Preliminaries

In this section, we will precise our setting and notation by describing the graph model (the model for the communication network of agents) and the attack model (the model for data privacy).

**Graph model.** We model our network of agents by a graph $G = (V, E)$, where $V$ is the set of vertices ($v \in V$ is a vertex) and $E$ is the set of edges (each edge is a tuple of two vertices). We denote the order of the graph, i.e. $|V|$, by $n$. We denote the degree of a vertex $v$ by $d_v$. This way, we denote the degree sequence of $G$ by $\mathbf{d} = (d_1, \dots, d_n)$. Further, we denote the minimum degree and the maximum degree by $d_{\min}$ and $d_{\max}$, respectively.

We highlight that we will follow the notation where lower-case characters (e.g., $x$) indicate scalars or maps, bolded lower-case characters (e.g., $\mathbf{x}$) indicate vectors, bolded upper-case characters (e.g., $\mathbf{X}$) indicate matrices, and upper-case characters (e.g., $X$) do not have a strictly assigned role but they usually indicate random variables or sets. Also, for $x \in \mathbb{N}$, we define $[x]$ as $\{1, \dots, x\}$. For $k \in \mathbb{Z}$ and $\mathbf{x} \in \mathbb{R}^n$, we denote the $k$-th raw moment $\frac{1}{n} \sum_{i=1}^{n} x_i^k$ by $\mu_{x,k}$ (when $k = 1$, we denote it by $\mu_x$). Similarly, for $k, l \in \mathbb{Z}$ and $\mathbf{x}, \mathbf{y} \in \mathbb{R}^n$, we will denote $\frac{1}{n} \sum_{i=1}^{n} x_i^k y_i^l$ by $\mu_{x^k y^l}$. Finally, we denote the vector of 1's by $\mathbf{1}$. We provide the tables summarizing the notations in Section A.

Our graph $G$ is an undirected graph with no self-loops or multiple edges. We will model $G$ by a random graph (a graph with fixed vertices but the presence of the edges being determined by draws from probability distributions) drawn from the configuration model which is defined as follows:

**Definition 1.** *The configuration model is the probability distribution over graphs that is parametrized by a degree sequence $\mathbf{d}$, so that for $i \in [n-1]$ and $j \in \{i+1, \dots, n\}$, edge $(v_i, v_j)$ is present with probability*

$$\frac{d_i d_j}{\left(\sum_{i'=1}^{n} d_{i'}\right) - 1}.$$

**Attack model.** We assume that the agents are honest-but-curious, i.e., all agents follow the established protocols (they are honest), but they try to use the available information to infer sensitive information of other agents (they are curious). We remark that in our setting where agents publish information and can see published information from others but no other communication is possible, it is straightforward to protect against agents which are malicious in the sense they deviate from the protocol in order to obtain more information. Protecting against agents which deviate from the protocol to influence the result of the computation, e.g., by publishing false information (also called data poisoning), is out of the scope of this paper.

We remark that we interpret a basic dataset as a table with instances over rows and (scalar) attributes over its columns. We introduce a simplified version of the $(\epsilon, \delta)$-indistinguishability proposed by Dwork et al. (2006), commonly known as differential privacy:

**Definition 2.** *Let $\epsilon, \delta \geq 0$. A randomized algorithm $\mathcal{A}$ is $(\epsilon, \delta)$-differentially private if and only if for all tuples $(\mathcal{D}, \mathcal{D}')$ in a collection where datasets $\mathcal{D}$ and $\mathcal{D}'$ differ only in the attribute of one instance, and for all $\mathcal{S} \subseteq \text{image}(\mathcal{A})$ we have*

$$\Pr(\mathcal{A}(\mathcal{D}) \in \mathcal{S}) \leq e^{\epsilon} \Pr(\mathcal{A}(\mathcal{D}') \in \mathcal{S}) + \delta.$$

In local differential privacy, the common idea is to add noise to attributes. We refer to such attributes as sensitive attributes. In central differential privacy, central coordinators add noise

to statistics computed from attributes, thus it is more common to refer to sensitive statistics as opposed to sensitive attributes. In this work, we will focus on local differential privacy.

A classic strategy to guarantee $(\epsilon, \delta)$-differential privacy is to generate noisy values from the Gaussian mechanism. We define the Gaussian mechanism in the context of local differential privacy:

**Definition 3.** *Let $\epsilon, \delta > 0$. Let $X \subseteq \mathbb{R}$. Let $x \in X$ be a scalar attribute. Let $x_{\min}$ and $x_{\max}$ be, respectively, the smallest and the largest element in $X$. The Gaussian mechanism is a mechanism that privatizes $x$ by taking an observation of the following random variable:*

$$\mathcal{N}\left(x, \frac{2\log(1.25/\delta)\Delta_2^2(x)}{\epsilon^2}\right),$$

*where*

$$\Delta_2(x) = \|x_{\max} - x_{\min}\|_2 \tag{2.1}$$

*is the $l_2$ sensitivity. (The definition can be generalized to vector attributes.)*

We also give a definition of $(\epsilon, \delta)$-differential privacy for graph data (extends Definition 2):

**Definition 4.** *Let $\epsilon, \delta \geq 0$. A randomized algorithm $\mathcal{A}$ is $(\epsilon, \delta)$-differentially private if and only if for all triples $(\mathcal{D}, \mathcal{D}', v_o)$ datasets $\mathcal{D}, \mathcal{D}'$ differ only in vertex $v_o$ and its attributes (i.e., the difference is in the label value and the presence/absence of one edge), and for all $\mathcal{S} \subseteq \mathrm{image}(\mathcal{A})$ we have*

$$\Pr(\mathcal{A}(\mathcal{D}) \in \mathcal{S}) \leq e^\epsilon \Pr(\mathcal{A}(\mathcal{D}') \in \mathcal{S}) + \delta.$$

## 3 Literature study

We are not aware of another approach of distributed averaging without handshakes and privatized degrees to which we could directly compare our approach. This way, we will discuss some works with partial solutions and give some reference works that were taken as building blocks.

If the degree was not a sensitive attribute, our problem can be solved by a gossip algorithm that corrects the bias by an application of the Metropolis–Hastings algorithm (Hastings, 1970). More specifically, in every gossip iteration, an agent can make use of the degrees of its neighbors to correct the bias from every value that will be included in the average.

Further, if handshakes among agents were allowed, the community could solve the problem by making use of an agreed-upon overlay network (a network "built" on top of the initial one). Though, even if overlay networks were possible, such solution would be characterized by the trade-off between robustness and communication delay, whereas in this work we try to maintain both. For example, the spanning tree overlay network guarantees unbiased averages and requires only few handshakes though it is vulnerable to node failure.

Then, there are several gossip algorithms that solve our problem by relying on handshakes. Boyd et al. (2006) require the agents to agree on an independent edge set (so-called matching). Kempe et al. (2003) assume that every agent knows if a sent message failed to reach its destination. Bellet et al. (2019) assume that an agent always accepts a message sent to it. Dellenbach et al. (2018),

every two neighbors initialize their communication by sharing a value related to a positive noise for one and a negative noise for other. Ridgley et al. (2019) use pushing as opposed to pulling, which in some contexts results in a weaker notion of self-organization.

We remark that garbled circuits (Gascón et al., 2017; Nikolaenko et al., 2013) is a common alternative for gossip algorithms. However, they rely on public-key cryptography and the exchange of public keys is a form of handshakes.

Now we introduce the reference works that were taken as building blocks. Oliveira (2009) gave a theorem on the matrix norm between the adjacency matrix of an Erdős–Rényi random graph and its expectation. Such result is useful for obtaining guarantees for distributed computations on graphs modelled by the Erdős–Rényi random graph, and it has motivated us to work on establishing a similar groundwork for graph models with arbitrary degree sequences. Regarding other works, Iutzeler et al. (2013) provided analysis of a gossip algorithm for distributed averaging on graphs with arbitrary degree sequences. Chierichetti et al. (2011) considered graph models characterized by a power-law degree sequence. Bellet et al. (2019) provided a definition of differential privacy for gossip algorithms. Lindell and Pinkas (2009) provided a compendium on privacy-preserving distributed averaging techniques (though this work focuses more on security than data privacy). Aysal et al. (2009) considered a gossip algorithm that is asynchronous. Bell et al. (2020) applied unbiased averages that were computed in a distributed manner for fitting a regression model.

## 4  Approach

In this section, we will describe the simple gossip algorithm that enables distributed averaging in communication networks without central coordinators. Then, we will provide a bias removal mechanism so that the simple gossip algorithm can be applied for computing unbiased averages.

### 4.1  The simple gossip algorithm

Let $\circ$ in $\mathbf{d}^{\circ-1}$ denote the operator for the element-wise power. We define the simple gossip algorithm:

---
**Algorithm 1:** `SimpleGossip` (`SiGo`)

---
**Input** : $\mathbf{A} \in [0,1]^{n \times n}$ : adjacency matrix of the graph of agents
$\quad\quad\quad$ $\mathrm{it_{go}} \in \mathbb{N}$ : number of gossip iterations
$\quad\quad\quad$ $\mathbf{w} \in \mathbb{R}^n$
**Output:** $\mathbf{z} \in \mathbb{R}^n$
$\mathbf{d} \leftarrow \sum_{i=1}^{n} \mathbf{A}_{\cdot,i}$
$\mathbf{T} \leftarrow \mathrm{diag}(\mathbf{d}^{\circ-1})\mathbf{A}$
$\mathbf{z} \leftarrow \frac{1}{n}\mathbf{1}^{\mathrm{T}}\mathbf{T}^{\mathrm{it_{go}}}\mathbf{w}$

---

We remark that matrix $\mathbf{T}$ is the transition matrix and acts as an averaging operator. The construction of $\mathbf{T}$ involves the complete adjacency matrix $\mathbf{A}$ which suggests that the operation is synchronous. However, Boyd et al. (2006) mentions that such algorithm can be executed asynchronously (the involvement of $\mathbf{A}$ is partial in every iteration). We use the synchronous version for simplicity.

For Algorithm 1 to converge (elements of $\mathbf{z}$ get close to each other), it is required that $\mathrm{it_{go}}$ is high enough and $G$ is a simple, connected graph with at least one odd cycle. Kermarrec and van Steen (2007) indicate that $\mathrm{it_{go}} = \lceil \log n \rceil$ is sufficient for SiGo to converge, when $\log n$ is approximately the diameter of $G$ and the degree sequence $\mathbf{d}$ is power-law. We state a theorem for the value of the output of the algorithm when the algorithm converges and prove it in Subsection B.1:

**Theorem 1.** *Let* $\mathbf{w} \in \mathbb{R}^n$ *(the theorem holds when this value is a random vector). For every* $j \in [n]$,

$$\mathrm{SiGo}\left((w_i)_{i=1}^n\right)_j \approx \frac{1}{n}\frac{1}{\mu_d}\sum_{i=1}^n d_i w_i, \tag{4.1}$$

*where* $\mathrm{SiGo}\left((.)_{i=1}^n\right)_j$ *is $j$-th element of the output of Algorithm 1.*

We remark that Theorem 1 is convenient to use for graphs modelled by the configuration model (Definition 1) since their expected degree sequence is the degree sequence that parametrizes the configuration model. We conclude that the theorem shows that the resulting average is biased by $\mu_d$ and $d_i$ (for every $i \in [n]$) when $d_i \neq \mu_d$ (i.e. the graph is non-regular). To give an example, we will show that the squared difference between the output of Algorithm 1 and the average $\mu_w$ over the elements of $\mathbf{w}$ is non-zero. For $j \in [n]$, we have

$$\left(\mathrm{SiGo}\left((w_i)_{i=1}^n\right)_j - \mu_w\right)^2 \approx \left(\frac{1}{n}\frac{1}{\mu_d}\sum_{i=1}^n d_i w_i - \mu_w\right)^2 \qquad \text{(by Theorem 1)}$$

$$= \left(\frac{\mu_{dw}}{\mu_d} - \mu_w\right)^2, \tag{4.2}$$

which is not equal to 0 when the graph is non-regular, as only then $\mu_{dw} = \mu_d \mu_w$.

## 4.2 The bias removal mechanism

We will devise a bias removal mechanism to correct the bias illustrated by Equation 4.2.

We give an example of how the community can compute an unbiased estimate of $\mu_w$ using several runs of Algorithm 1. Firstly, the community performs two runs of Algorithm 1 (executed sequentially or in parallel), with inputs $(w_i d_i^{-1})_{i=1}^n$ and $(d_i^{-1})_{i=1}^n$, resulting in

$$\mathrm{SiGo}\left((w_i d_i^{-1})_{i=1}^n\right)_j \approx \frac{1}{n}\frac{1}{\mu_d}\sum_{i=1}^n w_i = \frac{\mu_w}{\mu_d}, \qquad \text{(by Theorem 1)}$$

$$\mathrm{SiGo}\left((d_i^{-1})_{i=1}^n\right)_j \approx \frac{1}{n}\frac{1}{\mu_d}\sum_{i=1}^n 1 = \frac{1}{\mu_d}, \qquad \text{(by Theorem 1)}$$

where $j \in [n]$. Then, the community can compute

$$\frac{\mathrm{SiGo}\left((w_i d_i^{-1})_{i=1}^n\right)_j}{\mathrm{SiGo}\left((d_i^{-1})_{i=1}^n\right)_j} \approx \mu_w. \tag{4.3}$$

Let $k \in \mathbb{Z}$ and let $j \in [n]$. We generalize Equation 4.3:

$$\frac{\texttt{SiGo}\left((w_i^k d_i^{-1})_{i=1}^n\right)_j}{\texttt{SiGo}\left((d_i^{-1})_{i=1}^n\right)_j} \approx \frac{\frac{1}{n}\frac{1}{\mu_d}\sum_{i=1}^n d_i(w_i^k d_i^{-1})}{\frac{1}{n}\frac{1}{\mu_d}\sum_{i=1}^n d_i(d_i^{-1})} \qquad \text{(by Theorem 1)} \qquad (4.4)$$

$$= \frac{1}{n}\sum_{i=1}^n w_i^k = \mu_{w^k}.$$

Further, we define U-statistics which generalizes the notion of unbiased estimates:

**Definition 5.** *Let $r \geq 1$ be a natural number. Let $n \geq r$ be a natural number. For $i \in [n]$, let $x_i \in \mathbb{R}^r$ be an observation of a random vector. Let $\phi : \mathbb{R}^r \to \mathbb{R}$. The value*

$$\frac{1}{\binom{n}{r}}\sum_{i=1}^{\binom{n}{r}} \phi(x_{i,1},\ldots,x_{i,r})$$

*is a U-statistic of degree $r$ and kernel $\phi$.*

We conclude that the estimates obtained from the bias removal mechanism (Equation 4.4) are U-statistics of degree 1 and kernel $\phi : x \mapsto x^k$ for $k \in \mathbb{Z}$. We remark that U-statistics of degree 1 are present in some machine learning applications, for example, in classic strategies for fitting linear regression models and bootstrap aggregation in random forests.

## 5 Use case on linear regression

We will show a use case for applying our approach for fitting a linear regression model, where every agent is attributed a sensitive individual value which is derived from their sensitive degrees, and every agent learns the regression model of those individual values.

We start by defining the simple linear regression model (with one feature and one-dimensional target value) of our use case. For every $i \in [n]$,

$$y_i = \theta_0 + \theta_1 x_i + \xi_{\text{reg},i}, \qquad (5.1)$$

where $\theta_0, \theta_1 \in \mathbb{R}$ are regression parameters, $\Xi_{\text{reg},i} \sim \text{uni}[-l_{\text{reg}}/2, l_{\text{reg}}/2]$ is independent, mean-0 regression noise, $l_{\text{reg}}$ is the length of the interval of the regression noise, $\xi_{\text{reg},i}$ is an observation of $\Xi_{\text{reg},i}$, $x_i = (d_i - \mu_d)^2$ are features, and $y_i$ are target values.

In regression, a common way to estimate the regression parameters $\theta_0, \theta_1$ is to perform computations from pairs $(y_i, x_i)$. We denote the estimates of $\theta_0, \theta_1$ by $\hat{\theta}_0, \hat{\theta}_1$, respectively.

Now we will show a basic strategy to obtain the estimates of the regression parameters from unbiased averages. Let $\mathbf{X} = \begin{bmatrix} \mathbf{1} & \mathbf{x} \end{bmatrix} \in \mathbb{R}^{n \times 2}$, where $\mathbf{x}$ is the vector of features (over all agents). Let $\mathbf{y}$ denote the vector of target values. Let $\hat{\theta}$ denote the vector of the estimates of the regression parameters.

The regression model (Equation 5.1) can be expressed and rearranged as follows:

$$\mathbf{y} = \mathbf{X}\hat{\theta} \iff \mathbf{X}^{\mathrm{T}}\mathbf{y} = \mathbf{X}^{\mathrm{T}}\mathbf{X}\hat{\theta}$$
$$\iff \hat{\theta} = (\mathbf{X}^{\mathrm{T}}\mathbf{X})^{-1}\mathbf{X}^{\mathrm{T}}\mathbf{y}$$
$$\iff \hat{\theta} = \left(\frac{1}{n}\mathbf{X}^{\mathrm{T}}\mathbf{X}\right)^{-1}\frac{1}{n}\mathbf{X}^{\mathrm{T}}\mathbf{y}, \tag{5.2}$$

since $\left(\frac{1}{n}\mathbf{X}^{\mathrm{T}}\mathbf{X}\right)^{-1} = n\left(\mathbf{X}^{\mathrm{T}}\mathbf{X}\right)^{-1}$. We have that

$$\frac{1}{n}\mathbf{X}^{\mathrm{T}}\mathbf{X} = \begin{bmatrix} 1 & \frac{1}{n}\sum_i x_i \\ \frac{1}{n}\sum_i x_i & \frac{1}{n}\sum_i x_i^2 \end{bmatrix} = \begin{bmatrix} 1 & \mu_x \\ \mu_x & \mu_{x^2} \end{bmatrix}, \tag{5.3}$$

$$\frac{1}{n}\mathbf{X}^{\mathrm{T}}\mathbf{y} = \begin{bmatrix} \frac{1}{n}\sum_i y_i & \frac{1}{n}\sum_i y_i x_i \end{bmatrix} = \begin{bmatrix} \mu_y & \mu_{yx} \end{bmatrix}, \tag{5.4}$$

thus the estimates of the regression parameters can be computed from the values $\mu_x, \mu_{x^2}, \mu_y$ and $\mu_{yx}$. By Definition 5, the values $\mu_x, \mu_{x^2}, \mu_y$ and $\mu_{yx}$ are U-statistics of degree 1.

Using Algorithm 1 and its bias removal mechanism (Equation 4.4), the values $\mu_x, \mu_{x^2}, \mu_y, \mu_{yx}$ can be computed from $\mathtt{SiGo}\left((d_i^{-1})_{i=1}^n\right)_j$, $\mathtt{SiGo}\left((x_i d_i^{-1})_{i=1}^n\right)_j$, $\mathtt{SiGo}\left((x_i^2 d_i^{-1})_{i=1}^n\right)_j$, $\mathtt{SiGo}\left((y_i d_i^{-1})_{i=1}^n\right)_j$, $\mathtt{SiGo}\left((y_i x_i d_i^{-1})_{i=1}^n\right)_j$, where $j \in \mathbb{N}$. Though we leave a remark that the community needs to firstly compute $\mu_d \approx \frac{1}{\mathtt{SiGo}\left((d_i^{-1})_{i=1}^n\right)_j}$ because for executing Algorithm 1 with some of the other inputs, it is required to firstly compute the features $x_i = (d_i - \mu_d)^2$ which involves $\mu_d$.

We will elaborate on the strategy which can be used by every agent for keeping their sensitive attributes differentially private. Firstly, we split our privacy budget $(\epsilon, \delta)$ evenly in five parts (i.e., $\epsilon/5, \delta/5$) because the community executes Algorithm 1 with five different sensitive attributes (as suggested in the previous paragraph, these are $d_i^{-1}$, $x_i d_i^{-1}$, $x_i^2 d_i^{-1}$, $y_i d_i^{-1}$ and $y_i x_i d_i^{-1}$) to obtain the U-statistics for regression, and thus we hide every input under independent and appropriate noise. Such even split is suboptimal because some of those inputs are correlated, though this aspect is outside the scope of this work. We remark that the even split is appropriate for the Gaussian mechanism (Definition 3) as suggested by the textbook on differential privacy by Dwork et al. (2014) Further, instead of adding noise to the inputs of Algorithm 1, we could split the privacy budget to two parts and hide only the attributes $d_i$ and the target values $y_i$. However, in the end we would not necessarily have significantly more accurate estimations of the U-statistics as the computations of the features amplify the noise. Also, such strategy would complicate the theoretical guarantees derived later.

We will define the privatized inputs to Algorithm 1. Let $k, l, m \in \mathbb{Z}$. A privatized input $\nu_{i,(k,l,m)}$ (with respect to the sensitive attribute $y_i^k x_i^l d_i^m$) is an observation of the following random variable:

$$\Xi_{\mathrm{dp},i,(k,l,m)} \sim \mathcal{N}\left(y_i^k x_i^l d_i^m, \sigma_{\mathrm{dp},(k,l,m)}^2\right), \tag{5.5}$$

where

$$\sigma_{\mathrm{dp},(k,l,m)}^2 = \frac{2\log(1.25/(\delta/5))\Delta_2^2(y_i^k x_i^l d_i^m)}{(\epsilon/5)^2} \qquad \text{(by Definition 3)}$$
$$= \frac{50\log\left(\frac{25}{4}\frac{1}{\delta}\right)\Delta_2^2(y_i^k x_i^l d_i^m)}{\epsilon}. \tag{5.6}$$

We have derived the $l_2$ sensitivities (Equation 2.1) required for regression in Subsection C.1. Further, we have described the clipping of the five privatized inputs needed for regression in Subsection C.2.

## 6 Error analysis

In this section, we will discuss the total theoretical error and the total empirical empirical error between an average of privatized attributes (Equation 5.5) computed by Algorithm 1 and its bias removal mechanism (Equation 4.4) and an average computed centrally and without privatization.

Firstly, for $k, l \in \mathbb{Z}$, we define the unbiased averages computed by our approach from privatized attributes as follows:

$$S_{(k,l)}^{\text{sa,dp,go}} = \frac{\texttt{SiGo}\left(\left(\Xi_{\text{dp},i,(k,l,-1)}\right)_{i=1}^{n}\right)_j}{\texttt{SiGo}\left(\left(\Xi_{\text{dp},i,(0,0,-1)}\right)_{i=1}^{n}\right)_j}, \qquad \text{(similarly as in Equation 4.4),} \qquad (6.1)$$

where $\Xi_{\text{dp},i,(k,l,-1)}$ is defined in Equation 5.5 and $j \in [n]$ is chosen arbitrarily because we assume that Algorithm 1 is run for enough iterations to converge. We remark that the superscripts in $S_{(k,l)}^{\text{sa,dp,go}}$ indicate the presence of the three components that contribute to the error: "sa" stands for finite sampling of individual values, "dp" stands for differential privacy for sensitive attributes and "go" stands for the application of Algorithm 1 and its bias removal mechanism.

Further, we define the averages computed centrally, without sampling of individual values and without privatization as follows:

$$S_{(k,l)} = \frac{1}{n} \sum_{i=1}^{n} Y_{\text{reg},i}^{k} x_i^l, \qquad (6.2)$$

where, for $i \in [n]$, $Y_{\text{reg},i}$ is a random variable with

$$\mathbb{E}[Y_{\text{reg},i}] = y_i, \qquad (6.3)$$

$$\text{var}\left(Y_{\text{reg},i}\right) = \text{var}\left(\Xi_{\text{reg},i}\right) = \frac{l_{\text{reg}}^2}{12}. \qquad (6.4)$$

**Theoretical error.** Now we define the total theoretical error as follows:

$$e_{(k,l)}^{\text{theo,total}} = \mathbb{E}\left[\left(S_{(k,l)} - S_{(k,l)}^{\text{sa,dp,go}}\right)^2\right],$$

where $S_{(k,l)}^{\text{sa,dp,go}}$ is defined in Equation 6.1 and $S_{(k,l)}$ is defined in Equation 6.2.

We will state a theorem on the asymptotic theoretical guarantees for the total theoretical error in the estimates required for regression.

**Theorem 2.** *For $(k, l) \in \{(0, 1), (0, 2), (1, 0), (1, 1)\}$, we have*

$$e_{(k,l)}^{theo,total} = \mathcal{O}\left(\frac{1}{n}\right),$$

*when the privacy budget $(\epsilon, \delta)$ is not extremely low and not extremely high.*

We remark that in Theorem 2 we have $(k, l) \in \{(0, 1), (0, 2), (1, 0), (1, 1)\}$ since for fitting a linear regression model we aim for the estimates $\hat{\mu}_x$, $\hat{\mu}_{x^2}$, $\hat{\mu}_y$, $\hat{\mu}_{yx}$ as suggested by Equations 5.3, 5.4. Thus, $e_{(0,1)}^{\text{theo,total}}$ is related to $\hat{\mu}_x$, $e_{(0,2)}^{\text{theo,total}}$ is related to $\hat{\mu}_{x^2}$, $e_{(1,0)}^{\text{theo,total}}$ is related to $\hat{\mu}_y$, and $e_{(1,1)}^{\text{theo,total}}$ is related to $\hat{\mu}_{yx}$.

We will sketch the proof of Theorem 2 by decomposing the total error into three components: the error due to sampling, the error due to differential privacy noise, and the error due to Algorithm 1 and its bias removal mechanism. We remark that for error decomposition, we have followed the lecture notes by Rosenberg (2016). We start by applying the triangle inequality as follows:

$$
\begin{aligned}
e_{(k,l)}^{\text{theo,total}} &= \mathbb{E}\left[\left(S_{(k,l)} - S_{(k,l)}^{\text{sa,dp,go}}\right)^2\right] \\
&\leq \mathbb{E}\left[\left(S_{(k,l)} - s_{(k,l)}^{\text{sa}}\right)^2\right] + \mathbb{E}\left[\left(s_{(k,l)}^{\text{sa}} - S_{(k,l)}^{\text{sa,dp}}\right)^2\right] + \mathbb{E}\left[\left(S_{(k,l)}^{\text{sa,dp}} - S_{(k,l)}^{\text{sa,dp,go}}\right)^2\right],
\end{aligned} \tag{6.5}
$$

where

$$
s_{(k,l)}^{\text{sa}} = \frac{1}{n}\sum_{i=1}^{n} y_i^k x_i^l, \tag{6.6}
$$

$$
S_{(k,l)}^{\text{sa,dp}} = \frac{1}{n}\sum_{i=1}^{n} \Xi_{\text{dp},i,(k,l,0)}, \tag{6.7}
$$

$\Xi_{\text{dp},i,(k,l,0)}$ is defined in Equation 5.5, $S_{(k,l)}$ is defined in Equation 6.2, $S_{(k,l)}^{\text{sa,dp,go}}$ is defined in Equation 6.1, and $(k, l) \in \{(0, 1), (0, 2), (1, 0), (1, 1)\}$ as required for regression (Equations 5.3, 5.4). We proceed with defining the theoretical error due to sampling by

$$
e_{(k,l)}^{\text{theo,sa}} = \mathbb{E}\left[\left(S_{(k,l)} - s_{(k,l)}^{\text{sa}}\right)^2\right]. \tag{6.8}
$$

In Subsection B.2, we show that

$$
e_{(k,l)}^{\text{theo,sa}} = \begin{cases} \frac{1}{n}\frac{l_{\text{reg}}^2}{12}\mu_{x^{2l}} & \text{when } k = 1, \\ 0 & \text{when } k = 0. \end{cases} \tag{6.9}
$$

Then, we define the theoretical error due to differential privacy by

$$
e_{(k,l)}^{\text{theo,dp}} = \mathbb{E}\left[\left(s_{(k,l)}^{\text{sa}} - S_{(k,l)}^{\text{sa,dp}}\right)^2\right]. \tag{6.10}
$$

In Subsection B.3, we show that

$$
e_{(k,l)}^{\text{theo,dp}} \approx \frac{1}{n}\tilde{\sigma}_{\text{dp},(k,l,0)}^2, \tag{6.11}
$$

where the clipped standard deviation $\tilde{\sigma}_{\text{dp},(k,l,0)}$ is defined in Equation B.10. Finally, we define the theoretical error due to Algorithm 1 and its bias removal mechanism by

$$
e_{(k,l)}^{\text{theo,go}} = \mathbb{E}\left[\left(S_{(k,l)}^{\text{sa,dp}} - S_{(k,l)}^{\text{sa,dp,go}}\right)^2\right]. \tag{6.12}
$$

In Subsection B.4, we show that

$$
e_{(k,l)}^{\text{theo,go}} \approx \frac{1}{n}\tilde{\sigma}_{\text{dp},(k,l,0)}^2 + \mu_{y^k x^l}^2 - 2\frac{\mu_{y^k x^l}^2}{\mu_d}\sqrt{n}\frac{\sqrt{2}\mu_d}{\sqrt{\mu_{d^2}}\tilde{\sigma}_{\text{dp},(0,0,-1)}}f_{\text{daw}}\left(\frac{\sqrt{n}}{\sqrt{2\mu_{d^2}}\tilde{\sigma}_{\text{dp},(0,0,-1)}}\right)
$$
$$
+ \left(\frac{1}{n}\frac{\mu_{d^2}}{\mu_d^2}\tilde{\sigma}_{\text{dp},(k,l,-1)}^2 + \frac{\mu_{y^k x^l}^2}{\mu_d^2}\right)n\frac{\mu_d^2}{\mu_{d^2}\tilde{\sigma}_{\text{dp},(0,0,-1)}^2}\left(\sqrt{n}\frac{\sqrt{2}}{\sqrt{\mu_{d^2}}\tilde{\sigma}_{\text{dp},(0,0,-1)}}f_{\text{daw}}\left(\frac{\sqrt{n}}{\sqrt{2\mu_{d^2}}\tilde{\sigma}_{\text{dp},(0,0,-1)}}\right) - 1\right),
$$

$$(6.13)$$

where $f_{\text{daw}}(x) = e^{-x^2}\int_0^x e^{t^2}\,dt$ and $\tilde{\sigma}_{\text{dp},(0,0,-1)}$ (Equation B.10) is neither close to 0 (privacy budget is extremely high) nor high (privacy budget is extremely low). In Subsections B.2, B.3, B.4 we conclude that $e_{(k,l)}^{\text{theo,sa}}$, $e_{(k,l)}^{\text{theo,dp}}$ and $e_{(k,l)}^{\text{theo,dp}}$ are $\mathcal{O}(\frac{1}{n})$, thus $e_{(k,l)}^{\text{theo,total}}$ is also $\mathcal{O}(\frac{1}{n})$.

**Empirical error.** We proceed with the definitions of the empirical errors in a similar way as was done for the theoretical errors. We define the empirical error due to sampling:

$$
e_{(k,l)}^{\text{emp,sa}} = \left(\frac{1}{n}\sum_{i=1}^n (y_i - \xi_{\text{reg},i})^k x_i^l - \frac{1}{n}\sum_{i=1}^n y_i^k x_i^l\right)^2,
$$

$$(6.14)$$

where the term $(y_i - \xi_{\text{reg},i})^k = (\theta_0 + \theta_1 x_i)^k$ and $(k,l) \in \{(0,1),(0,2),(1,0),(1,1)\}$ as required for regression (Equations 5.3, 5.4). Also, we define the empirical error due to differential privacy:

$$
e_{(k,l)}^{\text{emp,dp}} = \left(\frac{1}{n}\sum_{i=1}^n y_i^k x_i^l - \frac{1}{n}\sum_{i=1}^n \nu_{i,(k,l,0)}\right)^2.
$$

$$(6.15)$$

Then, we define the empirical error due to Algorithm 1 and its bias removal mechanism:

$$
e_{(k,l)}^{\text{emp,go}} = \left(\frac{1}{n}\sum_{i=1}^n \nu_{i,(k,l,0)} - \frac{\texttt{SiGo}\left((\nu_{i,(k,l,-1)})_{i=1}^n\right)_j}{\texttt{SiGo}\left((\nu_{i,(0,0,-1)})_{i=1}^n\right)_j}\right)^2,
$$

$$(6.16)$$

where $j \in [n]$ is chosen arbitrarily as for the theoretical error. Finally, we combine the three components to the total empirical error:

$$
e_{(k,l)}^{\text{emp,total}} = \left(\frac{1}{n}\sum_{i=1}^n (y_i - \xi_{\text{reg},i})^k x_i^l - \frac{\texttt{SiGo}\left((\nu_{i,(k,l,-1)})_{i=1}^n\right)_j}{\texttt{SiGo}\left((\nu_{i,(0,0,-1)})_{i=1}^n\right)_j}\right)^2.
$$

$$(6.17)$$

## 7 Experiments

In this section, we will specify our hypotheses and experiments, and interpret the results.

We have conducted two sets of experiments to verify the following hypotheses. The first hypothesis is that the total theoretical error (Equation 6.5) is close to the total empirical error (Equation 6.17). We state the details of the experiment for verifying it:

**Experiment 1.** *We will obtain the errors due to sampling (Equations 6.14, 6.9), the errors due to differential privacy (Equations 6.15, 6.11), the errors due to Algorithm 1 and its bias removal mechanism (Equations 6.16, 6.13), and the total errors (Equations 6.17, 6.5) for each estimate $\hat{\mu}_x$, $\hat{\mu}_{x^2}$, $\hat{\mu}_x$, $\hat{\mu}_{yx}$. The vertical axis will be the error scale, and the horizontal axis will indicate $n \in \{2^8, 2^8 + 2^7, 2^9, 2^9 + 2^8, 2^{10}, 2^{10} + 2^9\}$.*

The second hypothesis is that the regression parameters computed by Algorithm 1 and its bias removal mechanism lead to a lower mean squared error between the true target values and the predicted target values (over a test set) compared to the parameters computed by Algorithm 1 without its bias removal mechanism. We state the details of the experiment for verifying it:

**Experiment 2.** *We will compare the estimates $\hat{\theta}_0$, $\hat{\theta}_1$ (Equation 5.2) obtained using our approach to a baseline and a naive approach. The comparison will take the mean squared error between the true target values and and the predicted target values over a test set. In particular, in each experiment iteration, we will construct a test set by generating a power-law degree vector $d'$ of size $2^7$ (Subsection C.3) and generate a vector $y'_i$ (Equation 5.1). Then, we will predict $y'_{\text{pred},i} = \hat{\theta}_0 + \hat{\theta}_1(d'_i - \hat{\mu}_d)^2$ and compute the mean squared error $\frac{1}{2^7}\sum_{i=1}^{2^7}\left(y'_{\text{pred},i} - y'_i\right)^2$. For the baseline, we take the case when the estimates are computed centrally (Equation 6.6). For a naive approach, we take the case when the estimates are computed by Algorithm 1 and its bias removal mechanism is disabled. The privacy budget for the baseline and the naive approach is split in four even parts (the privacy budget in our approach is split in five even parts). The vertical axis will indicate the mean squared error, and the horizontal axis will indicate $\epsilon \in \{2^{-2}, 2^0, 2^2, 2^4, 2^6, 2^8\}$.*

We fix the number of experiment repetitions to $\text{it}_{\text{exp}} = 2^{11}$. Regarding randomization, in every experiment repetition we generate new features and target values (for synthetic graphs, we generate a new degree sequence and a new graph). In Subsection C.3, we discuss generation of synthetic graphs with power-law degree sequences. In Subsection C.4, we provide the values of the remaining experiments parameters. In Subsection C.5, we provide secondary hypotheses and experiments.

**The synthetic dataset.** In Experiment 1, the total theoretical error approaches the total empirical error, illustrated by Figure 1. (Here, we only interpret the total errors for the estimate $\hat{\mu}_{yx}$, i.e. $(k, l) = (1, 1)$. The remaining interpretations are moved to Subsection C.6). The difference between the two errors is mainly due to the approximation using the heuristics (Equations B.15, B.16).

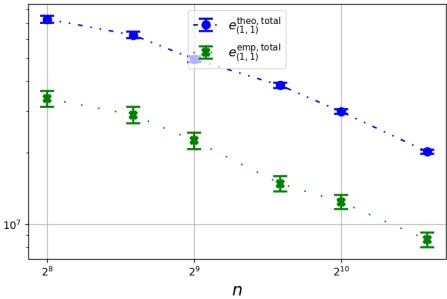

Figure 1: Comparison of the total errors (Equations 6.5, 6.17) for the estimate $\hat{\mu}_{yx}$, i.e. $(k, l) = (1, 1)$

In Experiment 2, the proposed approach performs better than the naive approach upon lower and higher privacy budget but not an intermediate one, illustrated in Figure 2. This suggests that the bias suggested by Equation 4.2 can get lower than the error due to differential privacy, as in the naive approach the privacy budget is split in four even parts as opposed to five.

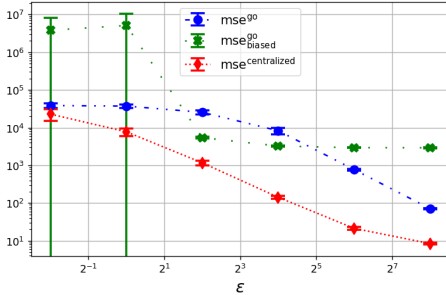

Figure 2: Comparison of our approach (Algorithm 1 and its bias removal mechanism) to the baseline (centralized averaging) and the naive approach (Algorithm 1 without its bias removal mechanism)

The experiments on the synthetic dataset were run on a home machine and took 1 hour and 43 minutes. The most time consuming operation was the matrix power (last line of Algorithm 1). The storage is mostly affected by the adjacency matrix of the graph. Our implementation becomes inappropriate to execute in practical time and ordinary memory when $n$ gets around $2^{13}$.

**The real datasets.** We consider the graphs of the email-Eu-core network dataset and the autonomous systems AS-733 dataset, both of which are part of SNAP (Leskovec and Krevl, 2014). The former graph has 1005 vertices, 25571 edges and its diameter is 7; and the latter graph has 6474 vertices, 13895 edges and its diameter is 9. We have processed the graphs by Step 3 of the procedure described in Subsection C.3 (this guarantees that Algorithm 1 converges and $d_{\max}$ stays not too high). We also remove the self-loops. Further, we fix the number of experiment repetitions to $\mathrm{it}_{\exp} = 2^7$ (opposed to $\mathrm{it}_{\exp} = 2^{11}$ since in the experiments on the real datasets we do not generate graphs).

We will interpret the results in Experiment 2. Algorithm 1 and its bias removal mechanism performs better than the naive approach, illustrated in Figure 3 Unlike for the synthetic dataset, Algorithm 1 with its bias removal mechanism performs better over all evaluated privacy budget values. This is the case because the average degree in the network is higher than for the synthetic dataset, which results in a higher bias.

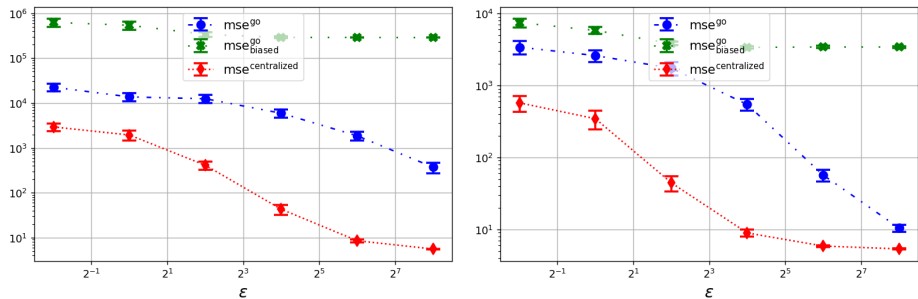

Figure 3: Comparing our approach (Algorithm 1 and its bias removal mechanism) to the baseline (centralized averaging) and the naive approach (Algorithm 1 without its bias removal mechanism). The email network dataset is on the left and the autonomous systems dataset is on the right

The experiment on the real datasets was run on a home machine and took 3 minutes for the email network dataset and 10 hours and 45 minutes on the autonomous systems dataset.

## 8 Conclusion

Algorithm 1 with its bias removal mechanism performs better than the naive approach for graphs with higher average degree partially because in such case the bias outweighs a lower split of the privacy budget (four rather than five parts). We remind that the even privacy split is suboptimal.

Further, the application of the Gaussian mechanism to privatize the degrees is suboptimal since degrees are natural numbers and the Gaussian mechanism produces real values. A more suitable strategy could be an application of a differential privacy mechanism that is appropriated for discrete values, e.g., the discrete Gaussian mechanism proposed by Canonne et al. (2020).

We remark that, in the regression model, features other than $x_i = (d_i - \mu_d)^2$ are possible. For this work, we chose $(d_i - \mu_d)^2$ because such choice guarantees the bias when the averages are computed by Algorithm 1 and it also leads to convenient clipping as $(d_i - \mu_d)^2$ is non-negative.

Finally, we remark that our approach might be applicable for estimating the unbiased sample variance which is a U-statistic of degree 2. That is, for $\mathbf{z} \in \mathbb{R}^n$, the unbiased sample variance is defined as $\frac{1}{n(n-1)} \sum_{j>i} (z_i - z_j)^2 = \frac{1}{n-1} \sum_i^n (z_i - \mu_z)^2$, and our approach can compute $\mu_z$ and $\frac{1}{n} \sum_i^n (z_i - \mu_z)^2$.

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

# A   Notation

Table 1: Summary of general notation

| Notation | Meaning | Comments |
|---|---|---|
| $x$ | lower-case character indicates a scalar | sometimes denotes a map to a scalar |
| $\mathbf{x}$ | bolded lower-case character indicates a vector | – |
| $\mathbf{X}$ | bolded upper-case character indicates a matrix | – |
| $X$ | upper-case character indicates a non-scalar | a random variable, a set, … |
| $x_i$ | $i$-th element of $\mathbf{x}$ | – |
| $[x]$ | $\{1, \ldots, x\}$, $x \in \mathbb{N}$ | – |
| $\mu_{x^k y^l}$ | $\frac{1}{n} \sum_{i=1}^{n} x_i^k y_i^l$, $k, l \in \mathbb{Z}$ | notation $\mu_{xy}$ implies $k = 1, l = 1$ |
| $\mathbf{1}$ | vector of 1's (of appropriate length) | – |

Table 2: Summary of notation related to graphs

| Notation | Meaning | Comments |
|---|---|---|
| $V$ | set of vertices | a vertex represents an agent |
| $E$ | set of edges | an edge represents a link between two agents |
| $G$ | graph | equivalent to a tuple $(V, E)$ |
| $n$ | number of vertices or order of a graph | equivalent to $|V|$ |
| $\mathbf{d}$ | $(d_1, \ldots, d_n)$ (degree sequence) | – |
| $d_{\min}$ | lowest degree in a degree sequence | – |
| $d_{\max}$ | highest degree in a degree sequence | – |

# B   Proofs

## B.1   Theorem 1

*Proof.* Our proof is based on a manipulation of the adjacency matrix which allows for a favorable eigendecomposition. Even though $\mathbf{T}$ is not symmetric, we can define

$$\mathbf{X} = \operatorname{diag}(\mathbf{d}^{\circ 1/2}) \mathbf{T} \operatorname{diag}(\mathbf{d}^{\circ -1/2}), \tag{B.1}$$

where $\circ$ denotes the operator for the element-wise power. We remark that $\mathbf{X}$ is symmetric. Both $\mathbf{T}$ and $\mathbf{X}$ have a largest eigenvalue 1, which has multiplicity 1 if $G$ is connected. The right eigenvector of $\mathbf{T}$ is $\mathbf{1}$, i.e., $\mathbf{1} = \mathbf{T1}$. It follows that

$$\operatorname{diag}(\mathbf{d}^{\circ 1/2}) \mathbf{1} = \mathbf{X} \operatorname{diag}(\mathbf{d}^{\circ 1/2}) \mathbf{1},$$

so $\operatorname{diag}(\mathbf{d}^{\circ 1/2}) \mathbf{1}$ is an eigenvector of $\mathbf{X}$ with eigenvalue 1, normalizing this eigenvector gives

$$\mathbf{v}_1 = \frac{\operatorname{diag}(\mathbf{d}^{\circ 1/2}) \mathbf{1}}{\sqrt{n \mu_d}}, \tag{B.2}$$

since $\sqrt{\sum_{i=1}^{n}(\sqrt{d_i})^2} = \sqrt{n\mu_d}$. We are interested in $\frac{1}{n}\mathbf{1}^{\mathrm{T}}\mathbf{T}^{\mathrm{it_{go}}}\mathbf{w}$. As $\mathbf{X}$ is symmetric, it has real eigenvalues and orthogonal eigenvectors. Let $\mathbf{X} = \mathbf{U}\mathbf{\Lambda}\mathbf{U}^{\mathrm{T}}$ be the eigenvalue decomposition of $\mathbf{X}$, where $\mathbf{\Lambda}$ is a diagonal matrix of eigenvalues in decreasing order, implying $\mathbf{\Lambda}_{1,1} = 1$. We can also see that $\mathbf{U}_{:,1} = \mathbf{v}_1$. For a sufficiently high $\mathrm{it_{go}}$, it holds that

$$
\begin{aligned}
\frac{1}{n}\mathbf{1}^{\mathrm{T}}\mathbf{T}^{\mathrm{it_{go}}}\mathbf{w} &= \frac{1}{n}\mathbf{1}^{\mathrm{T}}\left(\mathrm{diag}(\mathbf{d}^{\circ-1/2})\mathbf{X}\mathrm{diag}(\mathbf{d}^{\circ 1/2})\right)^{\mathrm{it_{go}}}\mathbf{w} && \text{(by Equation B.1)}\\
&= \frac{1}{n}\mathbf{1}^{\mathrm{T}}\mathrm{diag}(\mathbf{d}^{\circ-1/2})\mathbf{X}^{\mathrm{it_{go}}}\mathrm{diag}(\mathbf{d}^{\circ 1/2})\mathbf{w}\\
&= \frac{1}{n}\mathbf{1}^{\mathrm{T}}\mathrm{diag}(\mathbf{d}^{\circ-1/2})\mathbf{U}\mathbf{\Lambda}^{\mathrm{it_{go}}}\mathbf{U}^{\mathrm{T}}\mathrm{diag}(\mathbf{d}^{\circ 1/2})\mathbf{w}\\
&\approx \mathbf{1}^{\mathrm{T}}\mathrm{diag}(\mathbf{d}^{\circ-1/2})\mathbf{U}\mathrm{diag}(1,0,\ldots,0)\mathbf{U}^{\mathrm{T}}\mathrm{diag}(\mathbf{d}^{\circ 1/2})\mathbf{w}\\
&= \frac{1}{n}\mathbf{1}^{\mathrm{T}}\mathrm{diag}(\mathbf{d}^{\circ-1/2})\mathbf{v}_1\mathbf{v}_1^{\mathrm{T}}\mathrm{diag}(\mathbf{d}^{\circ 1/2})\mathbf{w}\\
&= \frac{1}{n}\mathbf{1}^{\mathrm{T}}\mathrm{diag}(\mathbf{d}^{\circ-1/2})\frac{\mathrm{diag}(\mathbf{d}^{\circ 1/2})\mathbf{1}}{\sqrt{n\mu_d}}\frac{\mathbf{1}^{\mathrm{T}}\mathrm{diag}(\mathbf{d}^{\circ 1/2})}{\sqrt{n\mu_d}}\mathrm{diag}(\mathbf{d}^{\circ 1/2})\mathbf{w} && \text{(by Equation B.2)}\\
&= \frac{1}{n\mu_d}\frac{1}{n}(\mathbf{1}^{\mathrm{T}}\mathbf{1})\mathbf{1}^{\mathrm{T}}\mathrm{diag}(\mathbf{d})\mathbf{w}\\
&= \frac{1}{n\mu_d}\mathbf{1}^{\mathrm{T}}\mathrm{diag}(\mathbf{d})\mathbf{w}\\
&= \frac{1}{n}\frac{1}{\mu_d}\sum_{i=1}^{n}d_i w_i.
\end{aligned}
$$

$\square$

## B.2 Error due to sampling

We will provide some guarantees for the theoretical error due to sampling. We firstly remark the true average $S_{(k,l)}$ defined in Equation 6.2 involves random variables only for the individual values $y_i$. The features $x_i$ are scalars because they only depend on the degrees $d_i$ and their mean $\mu_d$, and we define the regression model (Equation 5.1) when the graph of agents is already generated. In other words, the randomness in the graph generation will not be considered in the error due to sampling. For $(k,l) \in \{(0,1),(0,2),(1,0),(1,1)\}$ as required for regression, we have

$$
\begin{aligned}
e_{(k,l)}^{\mathrm{theo,sa}} &= \mathbb{E}\left[\left(S_{(k,l)} - s_{(k,l)}^{\mathrm{sa}}\right)^2\right] && \text{(by Equation 6.8)}\\
&= \mathbb{E}_{Y_{\mathrm{reg},i}}^{i=1,\ldots,n}\left[\left(\frac{1}{n}\sum_{i=1}^{n}Y_{\mathrm{reg},i}^k x_i^l - \frac{1}{n}\sum_{i=1}^{n}y_i^k x_i^l\right)^2\right] && \text{(by Equations 6.2, 6.6)}\\
&= \mathbb{E}\left[\left(S_{(k,l)} - \mu_{y^k x^l}\right)^2\right]\\
&= \mathbb{E}\left[S_{(k,l)}^2\right] - 2\mu_{y^k x^l}\mathbb{E}\left[S_{(k,l)}\right] + \mu_{y^k x^l}^2. && \text{(B.3)}
\end{aligned}
$$

We state a rearranged variance formula that we will use later:

$$
\mathbb{E}[Z^2] = \mathrm{var}\,(Z) + \mathbb{E}^2[Z], \tag{B.4}
$$

where $Z$ is any random variable. Further, for $k \in \{0, 1\}$, we derive

$$
\begin{aligned}
\mathbb{E}[S_{(k,l)}] &= \mathbb{E}\left[\frac{1}{n}\sum_{i=1}^{n} Y_{\text{reg},i}^{k} x_i^l\right] \\
&= \frac{1}{n}\sum_{i=1}^{n} \mathbb{E}\left[Y_{\text{reg},i}^{k}\right] x_i^l \\
&= \frac{1}{n}\sum_{i=1}^{n} y_i^k x_i^l \qquad\qquad\qquad \text{(by Equation 6.3)} \\
&= \mu_{y^k x^l}, \qquad\qquad\qquad\qquad\qquad\qquad\qquad\qquad\qquad \text{(B.5)}
\end{aligned}
$$

$$
\begin{aligned}
\text{var}\left(S_{(k,l)}\right) &= \text{var}\left(\frac{1}{n}\sum_{i=1}^{n} Y_{\text{reg},i}^{k} x_i^l\right) \\
&= \frac{1}{n^2}\text{var}\left(\sum_{i=1}^{n} Y_{\text{reg},i}^{k} x_i^l\right) \\
&= \frac{1}{n^2}\sum_{i=1}^{n} \text{var}\left(Y_{\text{reg},i}^{k} x_i^l\right) \\
&= \frac{1}{n^2}\sum_{i=1}^{n} x_i^{2l}\text{var}\left(Y_{\text{reg},i}^{k}\right) \\
&= \begin{cases} \frac{1}{n^2}\frac{l_{\text{reg}}^2}{12}\sum_{i=1}^{n} x_i^{2l} & \text{when } k = 1, \\ 0 & \text{when } k = 0, \end{cases} \qquad \text{(by Equation 6.4)} \\
&= \begin{cases} \frac{1}{n}\frac{l_{\text{reg}}^2}{12}\mu_{x^{2l}} & \text{when } k = 1, \\ 0 & \text{when } k = 0. \end{cases} \qquad\qquad\qquad\qquad\quad \text{(B.6)}
\end{aligned}
$$

Now we are ready to continue the derivation of the theoretical error due to sampling:

$$
\begin{aligned}
e_{(k,l)}^{\text{theo,sa}} &= \mathbb{E}\left[S_{(k,l)}^2\right] - 2\mu_{y^k x^l}\mathbb{E}\left[S_{(k,l)}\right] + \mu_{y^k x^l}^2 \qquad\qquad\qquad\qquad \text{(by Equation B.3)} \\
&= \text{var}\left(S_{(k,l)}\right) + \mathbb{E}^2[S_{(k,l)}] - 2\mu_{y^k x^l}\mathbb{E}\left[S_{(k,l)}\right] + \mu_{y^k x^l}^2 \qquad \text{(by Equation B.4)} \\
&= \begin{cases} \frac{1}{n}\frac{l_{\text{reg}}^2}{12}\mu_{x^{2l}} + \mu_{yx^l}^2 - 2\mu_{yx^l}\mu_{yx^l} + \mu_{yx^l}^2 & \text{when } k = 1, \\ \mu_{x^l}^2 - 2\mu_{x^l}^2 + \mu_{x^l}^2 & \text{when } k = 0, \end{cases} \quad \text{(by Equations B.5, B.6)} \\
&= \begin{cases} \frac{1}{n}\frac{l_{\text{reg}}^2}{12}\mu_{x^{2l}} & \text{when } k = 1, \\ 0 & \text{when } k = 0, \end{cases}
\end{aligned}
$$

We conclude that $e_{(0,l)}^{\text{theo,sa}} = 0$ and $e_{(1,l)}^{\text{theo,sa}} = \mathcal{O}(\frac{1}{n})$.

### B.3 Error due to differential privacy

We will provide some guarantees for the theoretical error due to differential privacy. For $(k, l) \in \{(0,1), (0,2), (1,0), (1,1)\}$ as required for regression, we have

$$
\begin{aligned}
e_{(k,l)}^{\text{theo,dp}} &= \mathbb{E}\left[\left(s_{(k,l)}^{\text{sa}} - S_{(k,l)}^{\text{sa,dp}}\right)^2\right] && \text{(by Equation 6.10)} \\
&= \mathbb{E}_{\Xi_{\text{dp},i,(k,l,0)}}^{i=1,\ldots,n}\left[\left(\frac{1}{n}\sum_{i=1}^{n} y_i^k x_i^l - \frac{1}{n}\sum_{i=1}^{n} \Xi_{\text{dp},i,(k,l,0)}\right)^2\right] && \text{(by Equations 6.6, 6.7)} \\
&= \mathbb{E}\left[\left(\mu_{y^k x^l} - S_{(k,l)}^{\text{sa,dp}}\right)^2\right] \\
&= \mu_{y^k x^l}^2 - 2\mu_{y^k x^l}\mathbb{E}\left[S_{(k,l)}^{\text{sa,dp}}\right] + \mathbb{E}\left[\left(S_{(k,l)}^{\text{sa,dp}}\right)^2\right]. && \text{(B.7)}
\end{aligned}
$$

We derive

$$
\begin{aligned}
\mathbb{E}[S_{(k,l)}^{\text{sa,dp}}] &= \mathbb{E}\left[\frac{1}{n}\sum_{i=1}^{n} \Xi_{\text{dp},i,(k,l,0)}\right] \\
&= \frac{1}{n}\sum_{i=1}^{n} \mathbb{E}\left[\Xi_{\text{dp},i,(k,l,0)}\right] \\
&= \frac{1}{n}\sum_{i=1}^{n} y_i^k x_i^l \\
&= \mu_{y^k x^l}, && \text{(B.8)}
\end{aligned}
$$

$$
\begin{aligned}
\text{var}\left(S_{(k,l)}^{\text{sa,dp}}\right) &= \text{var}\left(\frac{1}{n}\sum_{i=1}^{n} \Xi_{\text{dp},i,(k,l,0)}\right) \\
&= \frac{1}{n^2}\text{var}\left(\sum_{i=1}^{n} \Xi_{\text{dp},i,(k,l,0)}\right) \\
&= \frac{1}{n^2}\sum_{i=1}^{n} \text{var}\left(\Xi_{\text{dp},i,(k,l,0)}\right) \\
&= \frac{1}{n}\sigma_{\text{dp},(k,l,0)}^2. && \text{(by Equation 5.6)} && \text{(B.9)}
\end{aligned}
$$

However, the variance $\sigma_{\text{dp},(k,l,m)}^2$ is larger than what we have in practice because we clip the sensitive attributes as discussed in Subsection C.2. Since we use the Gaussian mechanism where differential privacy noise comes from the normal distribution, we approximate the variance of the clipped privatized attributes using the formula for the variance of the truncated normal distribution. This way, for $i \in [n]$ and appropriate $k, l, m$, we have

$$
\tilde{\sigma}_{\text{dp},(k,l,m)}^2 = \sigma_{\text{dp},(k,l,m)}^2 \left(1 + \frac{\alpha\phi(\alpha) - \beta\phi(\beta)}{\omega(\beta) - \omega(\alpha)} - \left(\frac{\phi(\alpha) - \phi(\beta)}{\omega(\beta) - \omega(\alpha)}\right)^2\right), \tag{B.10}
$$

where $\alpha = \frac{a - \nu_{i,(k,l,m)}}{\sigma_{\mathrm{dp},(k,l,m)}}$, $\beta = \frac{b - \nu_{i,(k,l,m)}}{\sigma_{\mathrm{dp},(k,l,m)}}$, $a$ is the lowest value to which $\nu_{i,(k,l,m)}$ is clipped, $b$ is the highest value to which $\nu_{i,(k,l,m)}$ is clipped, $\phi$ is the probability density function of the standard normal distribution and $\omega$ is the cumulative distribution of the standard normal distribution.

We remark that we approximate the variance of the clipped privatized attributes that were generated from the Gaussian mechanism as opposed to generating the privatized attributes from the truncated normal distribution since we have not found the differential privacy guarantees for the truncated normal distribution.

Now we are ready to continue the derivation of the theoretical error due to differential privacy:

$$
\begin{aligned}
e_{(k,l)}^{\mathrm{theo,dp}} &= \mu_{y^k x^l}^2 - 2\mu_{y^k x^l} \mathbb{E}\left[ S_{(k,l)}^{\mathrm{sa,dp}} \right] + \mathbb{E}\left[ \left( S_{(k,l)}^{\mathrm{sa,dp}} \right)^2 \right] && \text{(by Equation B.7)} \\
&= \mu_{y^k x^l}^2 - 2\mu_{y^k x^l} \mathbb{E}\left[ S_{(k,l)}^{\mathrm{sa,dp}} \right] + \mathrm{var}\left( S_{(k,l)}^{\mathrm{sa,dp}} \right) + \mathbb{E}^2[S_{(k,l)}^{\mathrm{sa,dp}}] && \text{(by Equation B.4)} \\
&\approx \mu_{y^k x^l}^2 - 2\mu_{y^k x^l}^2 + \frac{1}{n}\tilde{\sigma}_{\mathrm{dp},(k,l,0)}^2 + \mu_{y^k x^l}^2 && \text{(by Equations B.10, B.8)} \\
&= \frac{1}{n}\tilde{\sigma}_{\mathrm{dp},(k,l,0)}^2.
\end{aligned}
$$

We conclude that $e_{(k,l)}^{\mathrm{theo,dp}} = \mathcal{O}(\frac{1}{n})$.

### B.4 Error due to Algorithm 1 and its bias removal mechanism

We will provide some guarantees for the theoretical error due to Algorithm 1 and its bias removal mechanism. For $(k, l) \in \{(0, 1), (0, 2), (1, 0), (1, 1)\}$ as required for regression, we have

$$e_{(k,l)}^{\text{theo,go}} = \mathbb{E}\left[\left(S_{(k,l)}^{\text{sa,dp}} - S_{(k,l)}^{\text{sa,dp,go}}\right)^2\right] \quad \text{(by Equation 6.12)}$$

$$= \mathbb{E}_{\substack{i=1,\ldots,n \\ \Xi_{\text{dp},i,(k,l,0)}, \\ \Xi_{\text{dp},i,(k,l,-1)}, \\ \Xi_{\text{dp},i,(0,0,-1)}}}\left[\left(\frac{1}{n}\sum_{i=1}^n \Xi_{\text{dp},i,(k,l,0)} - \frac{\texttt{SiGo}\left((\Xi_{\text{dp},i,(k,l,-1)})_{i=1}^n\right)_j}{\texttt{SiGo}\left((\Xi_{\text{dp},i,(0,0,-1)})_{i=1}^n\right)_j}\right)^2\right] \quad \text{(by Equations 6.7, 6.1)}$$

$$= \mathbb{E}\left[\left(S_{(k,l)}^{\text{sa,dp}}\right)^2\right] + \mathbb{E}\left[\left(S_{(k,l)}^{\text{sa,dp,go}}\right)^2\right] - 2\mathbb{E}\left[S_{(k,l)}^{\text{sa,dp}} S_{(k,l)}^{\text{sa,dp,go}}\right]$$

$$= \mathbb{E}\left[\left(S_{(k,l)}^{\text{sa,dp}}\right)^2\right]$$

$$+ \mathbb{E}\left[\left(\texttt{SiGo}\left((\Xi_{\text{dp},i,(k,l,-1)})_{i=1}^n\right)_j\right)^2\right] \mathbb{E}\left[\left(\frac{1}{\texttt{SiGo}\left((\Xi_{\text{dp},i,(0,0,-1)})_{i=1}^n\right)_j}\right)^2\right]$$

$$- 2\mathbb{E}\left[S_{(k,l)}^{\text{sa,dp}}\right] \mathbb{E}\left[\texttt{SiGo}\left((\Xi_{\text{dp},i,(k,l,-1)})_{i=1}^n\right)_j\right] \mathbb{E}\left[\frac{1}{\texttt{SiGo}\left((\Xi_{\text{dp},i,(0,0,-1)})_{i=1}^n\right)_j}\right]$$

$$\approx \mathbb{E}\left[\left(S_{(k,l)}^{\text{sa,dp}}\right)^2\right]$$

$$+ \mathbb{E}\left[\left(\frac{1}{n}\frac{1}{\mu_d}\sum_{i=1}^n d_i \Xi_{\text{dp},i,(k,l,-1)}\right)^2\right] \mathbb{E}\left[\left(\frac{1}{\frac{1}{n}\frac{1}{\mu_d}\sum_{i=1}^n d_i \Xi_{\text{dp},i,(0,0,-1)}}\right)^2\right] \quad \text{(by Theorem 1)}$$

$$- 2\mathbb{E}\left[S_{(k,l)}^{\text{sa,dp}}\right] \mathbb{E}\left[\frac{1}{n}\frac{1}{\mu_d}\sum_{i=1}^n d_i \Xi_{\text{dp},i,(k,l,-1)}\right] \mathbb{E}\left[\frac{1}{\frac{1}{n}\frac{1}{\mu_d}\sum_{i=1}^n d_i \Xi_{\text{dp},i,(0,0,-1)}}\right] \quad \text{(B.11)}$$

We express

$$\mathbb{E}\left[\frac{1}{n}\frac{1}{\mu_d}\sum_{i=1}^n d_i \Xi_{\text{dp},i,(k,l,-1)}\right] = \frac{1}{n}\frac{1}{\mu_d}\sum_{i=1}^n d_i \mathbb{E}\left[\Xi_{\text{dp},i,(k,l,-1)}\right]$$

$$= \frac{\mu_{y^k x^l}}{\mu_d}, \quad \text{(by Equation 5.5)} \quad \text{(B.12)}$$

$$\text{var}\left(\frac{1}{n}\frac{1}{\mu_d}\sum_{i=1}^n d_i \Xi_{\text{dp},i,(k,l,-1)}\right) = \frac{1}{n^2}\frac{1}{\mu_d^2}\sum_{i=1}^n d_i^2 \text{var}\left(\Xi_{\text{dp},i,(k,l,-1)}\right)$$

$$= \frac{1}{n}\frac{\mu_{d^2}}{\mu_d^2}\tilde{\sigma}_{\text{dp},(k,l,-1)}^2. \quad \text{(by Equation B.10)} \quad \text{(B.13)}$$

Thus,

$$
\begin{aligned}
\mathbb{E}&\left[\left(\frac{1}{n}\frac{1}{\mu_d}\sum_{i=1}^{n}d_i\Xi_{\mathrm{dp},i,(k,l,-1)}\right)^2\right]\\
&=\mathrm{var}\left(\frac{1}{n}\frac{1}{\mu_d}\sum_{i=1}^{n}d_i\Xi_{\mathrm{dp},i,(k,l,-1)}\right)+\mathbb{E}^2\left[\frac{1}{n}\frac{1}{\mu_d}\sum_{i=1}^{n}d_i\Xi_{\mathrm{dp},i,(k,l,-1)}\right] \qquad \text{(by Equations B.4)}\\
&=\frac{1}{n}\frac{\mu_{d^2}}{\mu_d^2}\tilde{\sigma}^2_{\mathrm{dp},(k,l,-1)}+\frac{\mu^2_{y^k x^l}}{\mu_d^2}. \qquad\qquad\qquad\qquad\qquad\qquad\qquad \text{(by Equations B.10, B.12)}
\end{aligned}
$$
$$(B.14)$$

We state some heuristics for a Gaussian variable $Z$ with mean $\mu$ and variance $\sigma^2$ ($\mu$ and $\sigma$ are not too close to 0 and $\sigma$ not too high):

$$
\mathbb{E}\left[\frac{1}{Z}\right]\approx\frac{\sqrt{2}}{\sigma}f_{\mathrm{daw}}\left(\frac{\mu}{\sqrt{2}\sigma}\right), \tag{B.15}
$$

$$
\mathbb{E}\left[\frac{1}{Z^2}\right]\approx\frac{1}{\sigma^2}\left(\mu\frac{\sqrt{2}}{\sigma}f_{\mathrm{daw}}\left(\frac{\mu}{\sqrt{2}\sigma}\right)-1\right), \tag{B.16}
$$

where $f_{\mathrm{daw}}(x)=e^{-x^2}\int_0^x e^{t^2}\,dt$ is known as the Dawson function, and the comparison of the heuristics to a sample of $\frac{1}{Z}$ is performed on *stats stack exchange* (linguisticturn , https://stats.stackexchange.com/users/328865/linguisticturn). This way,

$$
\begin{aligned}
\mathbb{E}&\left[\frac{1}{\frac{1}{n}\frac{1}{\mu_d}\sum_{i=1}^n d_i\Xi_{\mathrm{dp},i,(0,0,-1)}}\right]\approx\frac{\sqrt{2}}{\sqrt{\frac{1}{n}\frac{\mu_{d^2}}{\mu_d^2}\tilde{\sigma}^2_{\mathrm{dp},(0,0,-1)}}}f_{\mathrm{daw}}\left(\frac{\frac{1}{\mu_d}}{\sqrt{2}\sqrt{\frac{1}{n}\frac{\mu_{d^2}}{\mu_d^2}\tilde{\sigma}^2_{\mathrm{dp},(0,0,-1)}}}\right) \quad \text{(by Equations B.15, B.12, B.13)}\\
&=\sqrt{n}\frac{\sqrt{2}\mu_d}{\sqrt{\mu_{d^2}}\tilde{\sigma}_{\mathrm{dp},(0,0,-1)}}f_{\mathrm{daw}}\left(\frac{\sqrt{n}}{\sqrt{2\mu_{d^2}}\tilde{\sigma}_{\mathrm{dp},(0,0,-1)}}\right),
\end{aligned}
\tag{B.17}
$$

$$
\begin{aligned}
\mathbb{E}&\left[\left(\frac{1}{\frac{1}{n}\frac{1}{\mu_d}\sum_{i=1}^n d_i\Xi_{\mathrm{dp},i,(0,0,-1)}}\right)^2\right]\\
&\approx\frac{1}{\frac{1}{n}\frac{\mu_{d^2}}{\mu_d^2}\tilde{\sigma}^2_{\mathrm{dp},(0,0,-1)}}\left(\frac{1}{\mu_d}\frac{\sqrt{2}}{\sqrt{\frac{1}{n}\frac{\mu_{d^2}}{\mu_d^2}\tilde{\sigma}^2_{\mathrm{dp},(0,0,-1)}}}f_{\mathrm{daw}}\left(\frac{\frac{1}{\mu_d}}{\sqrt{2}\sqrt{\frac{1}{n}\frac{\mu_{d^2}}{\mu_d^2}\tilde{\sigma}^2_{\mathrm{dp},(0,0,-1)}}}\right)-1\right) \quad \text{(by Equations B.16, B.12, B.13)}\\
&=n\frac{\mu_d^2}{\mu_{d^2}\tilde{\sigma}^2_{\mathrm{dp},(0,0,-1)}}\left(\sqrt{n}\frac{\sqrt{2}}{\sqrt{\mu_{d^2}}\tilde{\sigma}_{\mathrm{dp},(0,0,-1)}}f_{\mathrm{daw}}\left(\frac{\sqrt{n}}{\sqrt{2\mu_{d^2}}\tilde{\sigma}_{\mathrm{dp},(0,0,-1)}}\right)-1\right).
\end{aligned}
\tag{B.18}
$$

We continue the derivation of the theoretical error due to Algorithm 1 and its bias removal mechanism from Equation B.11:

$$e_{(k,l)}^{\text{theo,go}} \approx \mathbb{E}\left[\left(S_{(k,l)}^{\text{sa,dp}}\right)^2\right] - 2\mathbb{E}\left[S_{(k,l)}^{\text{sa,dp}}\right]\mathbb{E}\left[\frac{1}{n}\frac{1}{\mu_d}\sum_{i=1}^n d_i\Xi_{\text{dp},i,(k,l,-1)}\right]\mathbb{E}\left[\frac{1}{\frac{1}{n}\frac{1}{\mu_d}\sum_{i=1}^n d_i\Xi_{\text{dp},i,(0,0,-1)}}\right]$$

$$+ \mathbb{E}\left[\left(\frac{1}{n}\frac{1}{\mu_d}\sum_{i=1}^n d_i\Xi_{\text{dp},i,(k,l,-1)}\right)^2\right]\mathbb{E}\left[\left(\frac{1}{\frac{1}{n}\frac{1}{\mu_d}\sum_{i=1}^n d_i\Xi_{\text{dp},i,(0,0,-1)}}\right)^2\right]$$

$$= \mathbb{E}\left[\left(S_{(k,l)}^{\text{sa,dp}}\right)^2\right] - 2\mathbb{E}\left[S_{(0,1)}^{\text{sa,dp}}\right]\frac{\mu_{y^k x^l}}{\mu_d}\sqrt{n}\frac{\sqrt{2}\mu_d}{\sqrt{\mu_{d^2}}\tilde{\sigma}_{\text{dp},(0,0,-1)}}f_{\text{daw}}\left(\frac{\sqrt{n}}{\sqrt{2\mu_{d^2}}\tilde{\sigma}_{\text{dp},(0,0,-1)}}\right) \text{ (by Equations B.12, B.17)}$$

$$+ \left(\frac{1}{n}\frac{\mu_{d^2}}{\mu_d^2}\tilde{\sigma}_{\text{dp},(k,l,-1)}^2 + \frac{\mu_{y^k x^l}^2}{\mu_d^2}\right)n\frac{\mu_d^2}{\mu_{d^2}\tilde{\sigma}_{\text{dp},(0,0,-1)}^2}\left(\sqrt{n}\frac{\sqrt{2}}{\sqrt{\mu_{d^2}}\tilde{\sigma}_{\text{dp},(0,0,-1)}}f_{\text{daw}}\left(\frac{\sqrt{n}}{\sqrt{2\mu_{d^2}}\tilde{\sigma}_{\text{dp},(0,0,-1)}}\right) - 1\right)$$

(by Equations B.14, B.18)

$$= \frac{1}{n}\tilde{\sigma}_{\text{dp},(k,l,0)}^2 + \mu_{y^k x^l}^2 - 2\frac{\mu_{y^k x^l}^2}{\mu_d}\sqrt{n}\frac{\sqrt{2}\mu_d}{\sqrt{\mu_{d^2}}\tilde{\sigma}_{\text{dp},(0,0,-1)}}f_{\text{daw}}\left(\frac{\sqrt{n}}{\sqrt{2\mu_{d^2}}\tilde{\sigma}_{\text{dp},(0,0,-1)}}\right) \text{ (by Equations B.4, B.9, B.8)}$$

$$+ \left(\frac{1}{n}\frac{\mu_{d^2}}{\mu_d^2}\tilde{\sigma}_{\text{dp},(k,l,-1)}^2 + \frac{\mu_{y^k x^l}^2}{\mu_d^2}\right)n\frac{\mu_d^2}{\mu_{d^2}\tilde{\sigma}_{\text{dp},(0,0,-1)}^2}\left(\sqrt{n}\frac{\sqrt{2}}{\sqrt{\mu_{d^2}}\tilde{\sigma}_{\text{dp},(0,0,-1)}}f_{\text{daw}}\left(\frac{\sqrt{n}}{\sqrt{2\mu_{d^2}}\tilde{\sigma}_{\text{dp},(0,0,-1)}}\right) - 1\right).$$

Since our analysis relies on the heuristics so that

$$\sqrt{n}\frac{\sqrt{2}\mu_d}{\sqrt{\mu_{d^2}}\tilde{\sigma}_{\text{dp},(0,0,-1)}}f_{\text{daw}}\left(\frac{\sqrt{n}}{\sqrt{2\mu_{d^2}}\tilde{\sigma}_{\text{dp},(0,0,-1)}}\right) \approx \mu_d$$

and

$$n\frac{\mu_d^2}{\mu_{d^2}\tilde{\sigma}_{\text{dp},(0,0,-1)}^2}\left(\sqrt{n}\frac{\sqrt{2}}{\sqrt{\mu_{d^2}}\tilde{\sigma}_{\text{dp},(0,0,-1)}}f_{\text{daw}}\left(\frac{\sqrt{n}}{\sqrt{2\mu_{d^2}}\tilde{\sigma}_{\text{dp},(0,0,-1)}}\right) - 1\right) \approx \mu_d^2 + \frac{1}{n}\tilde{\sigma}_{\text{dp},(k,l,-1)}^2,$$

we state a weak conclusion that $e_{(k,l)}^{\text{theo,go}} = \mathcal{O}(\frac{1}{n})$. We support this claim by indicating that $f_{\text{daw}}(\sqrt{n}) = \mathcal{O}(\frac{1}{\sqrt{n}})$ and

$$\left(\sqrt{n}\frac{\sqrt{2}}{\sqrt{\mu_{d^2}}\tilde{\sigma}_{\text{dp},(0,0,-1)}}f_{\text{daw}}\left(\frac{\sqrt{n}}{\sqrt{2\mu_{d^2}}\tilde{\sigma}_{\text{dp},(0,0,-1)}}\right) - 1\right) \approx \frac{\mu_d}{\mu_d} - 1 = 0,$$

which result in $e_{(k,l)}^{\text{theo,go}} = \mathcal{O}(\frac{1}{n})$.

## C  Secondary material

### C.1  Sensitivity derivations

We will use the Gaussian mechanism for generating differential privacy noise, thus will derive the sensitivity terms for the five sensitive attributes. We note that the sensitivity terms 2.1 involves the identity functions because we are concerned with local differential privacy. We give a list of individual descriptions:

- The sensitive attribute $d_i^{-1}$. We assume that $d_{\min} = 3$. Based on Definition 4, adjacents datasets of graph data differ in 1 edge, thus

$$
\begin{aligned}
\Delta_2(d^{-1}) &= \underset{d' \in [d_{\min}, d_{\max}]}{\arg\max} \left\| \frac{1}{d'} - \frac{1}{d'+1} \right\|_2 \\
&= \left\| \frac{1}{d_{\min}(d_{\min}+1)} \right\|_2 \\
&= \frac{1}{12}
\end{aligned}
$$

- The sensitive attribute $x_i d_i^{-1}$. Firstly, we assume that $\Delta_2(xd^{-1}) \leq 2\Delta_2(x)\Delta_2(d^{-1})$ claiming that, for scalars $a \geq a' \geq 0$ and $b \geq b' \geq 0$, we have

$$
\begin{aligned}
(ab - a'b') &\leq 2ab + 2a'b' - a'b - ab' \\
&= 2(a - a')(b - b') \\
&\iff ab + 3a'b' \geq a'b + ab'.
\end{aligned}
$$

Then,

$$
\begin{aligned}
\Delta_2(x) &= \underset{d' \in [d_{\min}, d_{\max}]}{\arg\max} \left\| (d' - \mu_d)^2 - (d'+1-\mu_d)^2 \right\|_2 \\
&= \left\| (d_{\max} - \mu_d)^2 - (d_{\max}+1-\mu_d)^2 \right\|_2
\end{aligned}
$$

- The sensitive attribute $x_i^2 d_i^{-1}$. Similarly as for the sensitive attribute $x_i d_i^{-1}$, we assume that $\Delta_2(x^2 d^{-1}) \leq 2\Delta_2(x^2)\Delta_2(d^{-1})$, where

$$
\begin{aligned}
\Delta_2(x^2) &= \underset{d' \in [d_{\min}, d_{\max}]}{\arg\max} \left\| (d' - \mu_d)^4 - (d'+1-\mu_d)^4 \right\|_2 \\
&= \left\| (d_{\max} - \mu_d)^4 - (d_{\max}+1-\mu_d)^4 \right\|_2
\end{aligned}
$$

- The sensitive attribute $yd_i^{-1}$. Similarly as for the sensitive attribute $x_i d_i^{-1}$, we assume that $\Delta_2(yd^{-1}) \leq 2\Delta_2(y)\Delta_2(d^{-1})$. Then, we claim that

$$
\Delta_2(y) = l_{\mathrm{reg}},
$$

as that's the largest difference between two independent values of regression noise, and state that in our setting the agents know $l_{\mathrm{reg}}$ (otherwise the agents could not compute this sensitivity). We highlight that in our setting every agent $i$ knows its individual target value $y_i = \theta_0 + \theta_1 x_i + \xi_{\mathrm{reg},i}$ though not its individual components $\theta_0$, $\theta_1$, $\mu_d$ and $\xi_{\mathrm{reg},i}$. We assume that $\theta_1$ is sufficiently lower than $l_{\mathrm{reg}}$ so that the influence of the sensitive attribute $d_i$ in $\Delta_2(y)$ is insignificant (and thus absent in the calculation) as $\theta_1 x_i$ is already hidden under regression noise

- The sensitive attribute $y_i x_i d_i^{-1}$. Similarly as for the sensitive attribute $x_i d_i^{-1}$, we assume that $\Delta_2(yxd^{-1}) \leq 3\Delta_2(y)\Delta_2(x)\Delta_2(d^{-1})$, though here we have a coefficient 3 because we have three sensitivity terms as opposed to two

### C.2 Clipping of differentially private values

We describe the clipping of the five privatized inputs needed for regression: $\nu_{i,(0,0,-1)}$, $\nu_{i,(0,1,-1)}$, $\nu_{i,(0,2,-1)}$, $\nu_{i,(1,0,-1)}$, $\nu_{i,(1,1,-1)}$. Our sensitive attributes do not span over the set of real numbers, thus the privatized values should stay in the original domains as the sensitive attributes. We have that $d^{-1} \geq \frac{1}{d_{\max}}$ because we had fixed $d_{\max}$. Then, we have $x_i = (d_i - \mu_d)^2 \geq 0$ and $x_i^2 = (d_i - \mu_d)^4 \geq 0$ because squared real numbers are always non-negative. Finally, we assume that $y_i \geq 0$.

We will clip the privatized inputs at their bounds, though above we listed only the lower bounds of the elements of the sensitive attributes, and clipping them only at the lower bound would shift away (bias) the mean of the privatized attributes from the mean of the sensitive attributes. To avoid this shift, we fix upper bounds at the distance equal to the difference between a sensitive attribute and its lowers bound. This way, for $i \in \mathbb{N}$,

$$\frac{1}{d_{\max}} \leq \nu_{i,(0,0,-1)} \leq \nu_{i,(0,0,-1)} + \left( \nu_{i,(0,0,-1)} - \frac{1}{d_{\max}} \right) = 2\nu_{i,(0,0,-1)} - \frac{1}{d_{\max}},$$

$$0 \leq \nu_{i,(0,1,-1)} \leq 2\nu_{i,(0,1,-1)},$$

$$0 \leq \nu_{i,(0,2,-1)} \leq 2\nu_{i,(0,2,-1)},$$

$$0 \leq \nu_{i,(1,0,-1)} \leq 2\nu_{i,(1,0,-1)},$$

$$0 \leq \nu_{i,(1,1,-1)} \leq 2\nu_{i,(1,1,-1)}.$$

### C.3 Generation of graph datasets with power-law degree sequences

We describe the procedure that we followed to generate graphs with power-law degree sequences:

1. We generate a power-law degree sequence $\mathbf{d}' = (d_1', \ldots, d_n')$ whose every element is generated (drawn independently) from the following probability distribution:

$$f_{\mathrm{pow}}\left(d' \mid \gamma\right) = \frac{d'^{-\gamma}}{\sum_{d'=1}^{d_{\max}-3} d'^{-\gamma}},$$

where $\gamma > 1$ and the support of the distribution is $[1, d_{\max}-3]$. We remark that a power-law degree sequence is characterized by a higher proportion of vertices being attributed with lower degrees and a lower proportion of vertices being attributed with higher degrees

2. We generate a graph using the configuration model (Definition 1), parametrizing it with the degree sequence previously generated power-law degree sequence

3. For $i \in [n]$, we check if $d_i > d_{\max} - 3$ and if so remove arbitrary edges that involve vertex $i$ until $d_i = d_{\max} - 3$. Then, for $i \in [n-1]$, we add edge $(v_i, v_{i+1})$; and for $i = n$, we add edge $(v_n, v_1)$ so that there is at least one cycle in the graph, which guarantees that the graph is connected. If, for example, edge $(v_i, v_{i+1})$ was already present, we would try to add a subsequent edge that is absent, that is, we would check edges $(v_i, v_{i+2})$, $(v_i, v_{i+3})$, $\ldots$, $(v_i, v_{i-1})$. This guarantees that the degree of each vertex increases by 2. Further, if edge $(v_1, v_3)$ is absent, we add it also because this guarantees that there is at least one cycle of odd length in the graph, as the presence of edges $(v_1, v_2), (v_2, v_3)$ is assured by

the previously mentioned procedure that connects the graph. Finally, if the degrees of some vertices are still lower than 3, we add some arbitrary edges so that the degree of every vertex is at least 3. Since the resulting graph is connected and has at least one cycle of odd length, the principles of DeGroot learning suggest that the averages computed by Algorithm 1 will eventually involve the values of all agents in the network and thus the algorithm will converge. We denote the degree sequence that results after the addition and removal edges by $\mathbf{d} = (d_1, \ldots, d_n)$. We remark that $\min(\mathbf{d}) = d_{\min} = 3$ and $\max(\mathbf{d}) = d_{\max}$

### C.4 Remaining experiment parameters

We list the values to which we will fix the parameters of the experiment setting (unless indicated differently in experiment descriptions):

- The order of the graph: $n = 2^{10}$

- The differential privacy parameters: $(\epsilon, \delta) = (2^2, 2^{-7})$

- The number of gossip iterations: $\mathrm{it}_{\mathrm{go}} = 2^{10}$

- The significance level $\alpha_{\mathrm{ci}} = 0.05$. We will compute the confidence intervals based on Student's $t$-distribution. That is, for a vector $\mathbf{z}$ of length $\mathrm{it}_{\mathrm{exp}}$ and whose elements are scalars (e.g., mean squared errors), we have the confidence interval $[\mu_z - l_{\mathrm{ci}}/2, \mu_z + l_{\mathrm{ci}}/2]$, where $\mu_z$ and $\sigma_z$ are the sample mean and the unbiased sample standard deviation computed from the elements of $\mathbf{z}$,
$$l_{\mathrm{ci}} = 2q\left(1 - \frac{\alpha_{\mathrm{ci}}}{2} \mid \mathrm{it}_{\mathrm{exp}} - 1\right) \frac{\sigma_z}{\sqrt{\mathrm{it}_{\mathrm{exp}}}}$$
is the length of the confidence interval and $q$ is the quantile function of Student's $t$-distribution

- The true parameters of the regression model: $\theta_0 = 2^{12}$, $\theta_1 = 2^0$

- The length of the support of the uniform distribution for regression noise: $l_{\mathrm{reg}} = 2^3$

- The scale parameter of the probability distribution used to generate the degree sequence: $\gamma = 2$

- The highest degree: $d_{\max} = 2^6$

### C.5 Secondary hypotheses and experiments

We have conducted two sets of experiments to verify the following secondary hypotheses.

The third (in total) hypothesis is that larger graphs lead to more precise estimations of the regression parameters. We state the details of the experiment for verifying it:

**Experiment 3.** *Similarly as in Experiment 2, we will evaluate the estimates $\hat{\theta}_0$, $\hat{\theta}_1$ expressed in Equation 5.2 by computing the mean squared error between the true values and the predicted values over a test set. In the illustration, we will have a curve for each parameter $\gamma \in \{2, 3, 4\}$. The vertical axis will indicate the mean squared error, and the horizontal axis will indicate $n \in \{2^8, 2^8 + 2^7, 2^9, 2^9 + 2^8, 2^{10}, 2^{10} + 2^9\}$.*

The fourth hypothesis is that the number of gossip iterations approximately equal to the logarithm of the number of vertices of a graph with a power-law degree sequence is sufficient for Algorithm 1 to converge. We state the details of the experiment for verifying it:

**Experiment 4.** *We will evaluate the variance of the sample that contains the estimates $\hat{\theta}_1$ obtained over all agents. This variance indicates the convergence of Algorithm 1 because the computation of $\hat{\theta}_1$ involves all four estimates $\hat{\mu}_x$, $\hat{\mu}_{x^2}$, $\hat{\mu}_y$, $\hat{\mu}_{yx}$. In the illustration, we will have a curve for each parameter $\mathrm{it}_{\mathrm{go}} \in \{2^4 \approx \log 2^{10}, 2^{10}, 2^{11}\}$. The vertical axis will indicate the variance of the elements of $\hat{\theta}_1$, and the horizontal axis will indicate $n \in \{2^8, 2^8 + 2^7, 2^9, 2^9 + 2^8, 2^{10}, 2^{10} + 2^9\}$.*

We interpret the results on Experiment 3 on the synthetic dataset. The mean squared error decreases when the number of vertices increases, illustrated in Figure 4. The illustration also suggests that the decrease is smoother when the scale parameter $\gamma$ is lower. This happens because for higher $\gamma$ the degree sequence is likely to miss some higher values in the interval $[d_{\min}, d_{\max}]$, and thus the estimates $\hat{\theta}_0, \hat{\theta}_1$ will lead to a poor fit once the test set happens to include a feature computed using a degree value that was absent upon computing the estimates for the initial fit. However, for $\gamma = 2$ and lower $n$ values we still have wider confidence intervals which is also caused by a poor fit.

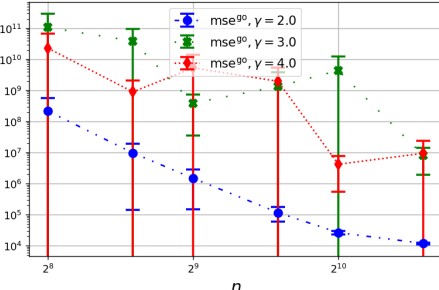

Figure 4: Comparison of the mean squared errors between the true values and the predicted values (over a test set) over several choices of $\gamma$

We interpret the results on Experiment 4 on the synthetic dataset. The variance of the sample that contains the estimates $\theta_1$ over all agents approaches a limit once $\mathrm{it}_{\mathrm{go}}$ reaches $2^{10}$, illustrated in Figure 5. As observed in Experiment 2, for lower $n$ values the confidence intervals are wider due to a poor fit. For this reason, we evaluate the convergence looking at higher $n$ values. When $\mathrm{it}_{\mathrm{go}} = 2^4$, the variance of the parameter estimates is already significantly low though not yet at the limit.

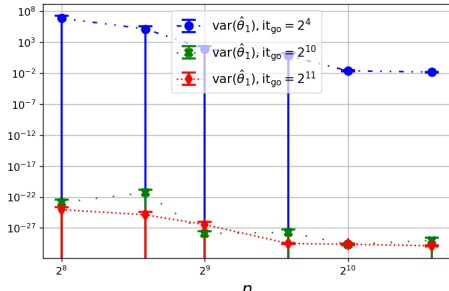

Figure 5: Comparison of the variance of the sample that contains the estimates $\hat{\theta}_1$ (over all agents) over several choices of $\text{it}_{\text{go}}$

The complete run of experiments on the synthetic dataset performed on a home machine took 5 hours and 45 minutes.

We interpret the results on Experiment 1 in on the real datasets. In Table 3, we provide the details of the convergence on the real graph datasets. We remark that $\text{it}_{\text{go}} = 2^4$ is not enough for Algorithm 1 to converge on the autonomous systems dataset. The better performance on the email network dataset can be explained by the presence of a lower diameter.

Table 3: Comparison of the variance of the sample that contains the estimates $\hat{\theta}_1$ (over all agents) over several choices of $\text{it}_{\text{go}}$

| $\text{it}_{\text{go}}$ | $\text{var}(\hat{\theta}_1)$ (email network) | $\text{var}(\hat{\theta}_1)$ (autonomous systems) |
|---|---|---|
| $2^4$ | $\approx 10^{-3}$ | $\approx 10^2$ |
| $2^{10}$ | $\approx 10^{-30}$ | $\approx 10^{-15}$ |
| $2^{11}$ | $\approx 10^{-30}$ | $\approx 10^{-28}$ |

The complete run of experiments on the real datasets performed on a home machine and took 5 minutes for the email network dataset and 19 hours and 58 minutes on the autonomous systems dataset.

### C.6 Continuation of Experiment 1 on the synthetic dataset

We will interpret the remaining results on Experiment 1. The theoretical errors for the estimates of the U-statistics required for regression approximate the empirical errors significantly closely, as illustrated by Figures 6, 7, 8, 9. In Figure 6, we observe that the theoretical error due to sampling is very close to the empirical error. In Figure 7, the theoretical error due to differential privacy is sometimes lower than the empirical error. This effect comes from the approximation (Equation B.10) of the variance of the clipped noisy values, which neglects the fact that a significant portion of clipped noisy values should be at the extremes of the interval (the shape of the probability distribution should have increasing curves at the boundaries as opposed to the drastic disappearance of the tails as in the truncated normal distribution). In Figure 8, the theoretical error due to Algorithm 1 and its bias removal mechanism is sometimes lower than the empirical error. This

effect comes from the same reason as for the error due to differential privacy noise and also due to the use of the approximation for the reciprocal of a random value obtained from the heuristics given in Equations B.15, B.16. The heuristics worsen when the mean of the random variable gets closer to 0, which, in our case, happens when $d_{\max}$ increases. The total error is illustrated in Figure 9.

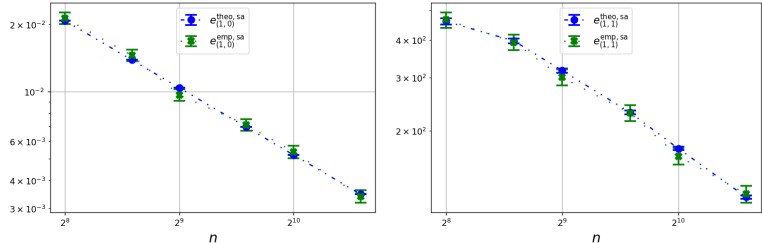

Figure 6: Comparison of the theoretical error and the empirical error due to sampling (Equations 6.8, 6.14) for the estimate $\hat{\mu}_y$, $\hat{\mu}_{yx}$

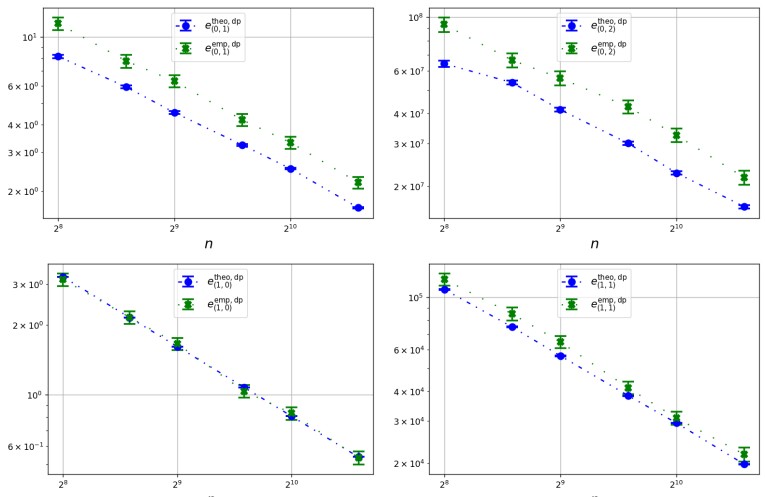

Figure 7: Comparison of the theoretical error and the empirical error due to differential privacy (Equations 6.10, 6.15) for the estimates $\hat{\mu}_x$, $\hat{\mu}_{x^2}$, $\hat{\mu}_y$, $\hat{\mu}_{yx}$

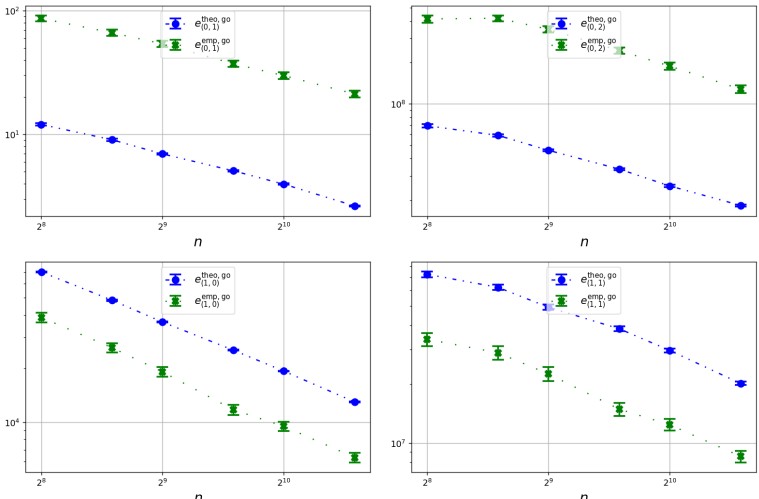

Figure 8: Comparison of the theoretical error and the empirical error due to Algorithm 1 and its bias removal mechanism (Equations 6.10, 6.15) for the estimates $\hat{\mu}_x$, $\hat{\mu}_{x^2}$, $\hat{\mu}_y$, $\hat{\mu}_{yx}$

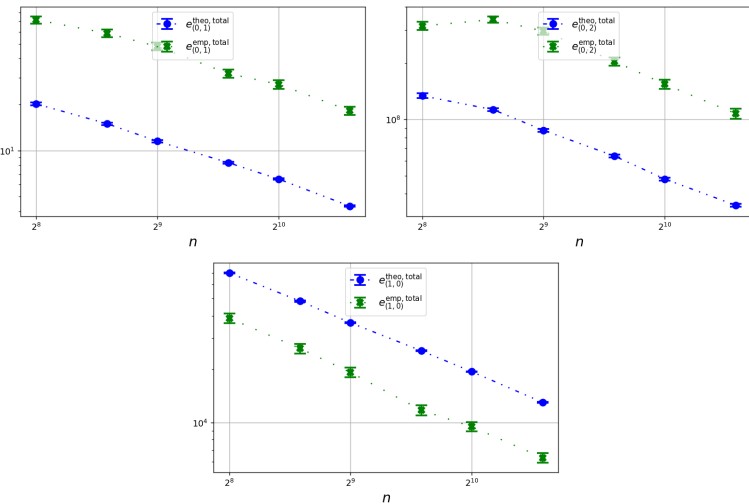

Figure 9: Comparison of the total errors (Equations 6.5, 6.17) for the estimates $\hat{\mu}_x$, $\hat{\mu}_{x^2}$, $\hat{\mu}_y$, $\hat{\mu}_{yx}$

