# OpenReview forum: "Estimating Unbiased Averages of Sensitive Attributes without Handshakes among Agents"
_TMLR — Withdrawn by Authors_

### Review · Reviewer_cPC8 · 2022-05-04

**Summary Of Contributions:**

This work investigates the estimation of averages and moments of data which is distributed among a large number of agents on a network, under privacy constraints where agents are 'honest but curious'. The main motivation of the paper is that consensus protocols typically use the degree of each node, which may itself lead to privacy violations. The paper considers a random graph model called the configuration model, which depends on a degree sequence and generates random edges that respect the degree sequence. The algorithm is based on a standard gossip algorithm, where the exchanged quantities are privatized by Gaussian noise addition. First, the paper presents the non-private version of the algorithm, together with a bias-removal method. Next, it discusses its use in a (1-dimensional) distributed linear regression example. Next it briefly discusses how the analysis and convergence may be affected by the privacy-preserving noise addition. Finally, numerical results on fictitious and real data are provided.

**Broader Impact Concerns:**

The most important concern I have about this work is that it might give the wrong impression that making $\epsilon=2^4$ (or even larger, as used in numerical results) is an acceptable choice for the privacy parameter. A similar concern applies to the choices of $\delta$.

**Requested Changes:**

The following changes I regard them as critical to consider the paper publishable.

1. As mentioned above, Theorem 1 needs a more rigorous statement about what is the accuracy guarantee obtained on the approximations for each agent, as well as a quantitative error bound on such accuracy measure. E.g., the issue of spectral gap and how it is controlled for the random graphs under consideration should be taken into account. BTW, it is rather unclear what is meant within parenthesis in this theorem.
2. For the linear regression application, there should be some a priori bound on $x,y$. This is briefly mentioned for $y$, but I don't see anywhere this being made explicit for $x$. Moreover, the sensitivity bounds in the Appendix are very unclear in this respect (in page 25, the sensitivity bounds on $x$ depend on the degree and on $\mu_d$, and I don't see how this relates to the covariate at all).
3. Theorem 2. Here again, something is said about an error bound when ``the privacy budget is not extremely low or extremely high.'' This needs to be made precise. Besides, the big-O is hiding factors that depend on these parameters, which I believe need to be made explicit.
4. Experiments in the main body of the paper do not explicit what the value of $\delta$ is, and the range of values of $\epsilon$ is (mostly) unacceptable. Keep in mind that for any $\epsilon>10$, the ratio of densities will only be bounded by $e^{10}$ (with probability $1-\delta$).


Here are some minor comments that I think would enhance the presentation of the paper:

1. There is no clear description about the problem being solved until page 5. This is not good for any reader trying to understand what the paper is about. What is missing is: What is the accuracy measure that will be used to quantify the success of the algorithm? Mean-squared error among the agents? Maximum error among agents? This also relates to the fact that Theorem 1 does not give any precise accuracy guarantee for what $\approx$ precisely means. Numerical experiments should Alsop be reporting the error on this accuracy measure.
2. Definition 3, and particularly $\Delta_2(x)$ gives the wrong impression that the amount of noise added might be dependent on $x$, whereas this noise must be worst-case over $x\in X$.
3. The proof of Theorem 1 seems to rely on the uniqueness of the leading eigenvector of the Laplacian. Even though the graph may be connected (which I suppose needs to be proved for the random graphs being used) its spectral gap can be ill-posed. This should affect convergence, yet this is not discussed at all in the paper. How this convergence is affected by the noise addition is also unclear.
4. Definition 5 is unclear. The index i runs from 1 to n, but the summation in the definition runs from 1 to $\binom{n}{r}$.

**Strengths And Weaknesses:**

Strengths:
1. The problem is interesting.
2. Based on the numerical results, their methods appear to work on very large-scale instances (although the resulting accuracy does not seem satisfactory)

Weaknesses:
1. The paper is unrigorous in various aspects. The convergence result for the consensus protocol does not give (asymptotic or non-asymptotic) precise accuracy guarantees, and just says the output is `close' (using $\approx$) to certain quantity. Other problems with the rigour on statements and analysis arise in the paper.
2. The accuracy of the methods on both real and fictitious data are too weak, for an application (linear regression with 2 regressors) which is too simple. The error in various figures have large orders of magnitude for any useful privacy parameter choice. On the other hand, for $(\epsilon,\delta)$-differential privacy to be really useful in practice, it is necessary that $\varepsilon \leq 10$ and $\delta\ll1/n$ (about the choice of $\delta$, see e.g. the name and shame mechanism example here https://dpcourse.github.io/lecnotes-web/lec-05-DP-II.pdf).

Considering all this, I believe this paper should not be accepted for publication in its current form, and (drastic) changes need to be made to it in order to become publishable.

---

### Review · Reviewer_YEWR · 2022-05-20

**Summary Of Contributions:**

I will be honest and start by saying that I have not worked previously in either differentially-private learning or this type of distributed learning scenarios. That said, I am afraid that I was not able to appreciate the contribution of the paper. More importantly, even after reading until page 5 and Sec. 4, I was still left wondering what the exact problem formulation is and why it is important. Unfortunately, the literature review section did not help me gain better understanding of the area, the particular problem that is being addressed and why it is important (perhaps through a concrete convincing application).

**Broader Impact Concerns:**

Given my non-expertise in the area, I have low confidence in my review given above. There is a chance I have missed an important point, otherwise I would suggest major revision/rewriting of the paper before re-review and potential publication.

**Requested Changes:**

I would like to see a concrete statement of the problem of interest early on together with its motivation and why we should study it (other than, generic statements such as "it has not been studies above")

**Strengths And Weaknesses:**

My main concern is already stated above. I also found the paper rather hard to read. The definitions and results (at least in Sec. 2 and 4) appear rather dry with little discussion on their interpretation and motivation. For example, what motivates definition 1 or definition 3? Are these well-known/standard? If so, please give references

---

### Review · Reviewer_VWQ2 · 2022-06-04

**Summary Of Contributions:**

The paper studies statistical tasks of mean-estimation and linear regression under the constraints of (i) multiple agents possessing independent samples, (ii) decentralized communications and (ii) differential privacy constraint on the private data of each agents. For both the statistical tasks, the paper proposes gossip based algorithms that provably achieve the desiderata while meeting the aforementioned constraints. A key observation in the paper is that a vanilla implementation of decentralized gossip based estimation is not consistent on all graphs - rather agents must perform bias correction to account for the degree heterogeneity. The degree heterogeneity arises due to the decentralized communication nature -- and agents that are more connected have better vantage point compared to agents with smaller degrees. Subsequently, the paper proposes simple mechanisms to account for this heterogeneity while simultaneously respecting differential privacy constraints.

**Requested Changes:**

Ensure all theorem/lemma/proposition statements rigorous.

It is ok to make an "informal theorem" in the main body. However, it is crucial to have a precise theorem/lemma statements at-least in the Appendix, so as to be able to parse the results. This is especially important to this paper as the main contributions are theoretical results.

**Strengths And Weaknesses:**

Strengths : Clear problem formulation and a stochastic model that is interesting and can provide a fertile ground to develop and benchmark decentralized, multi-agent, privacy preserving statistical estimation algorithms.

Weakness :

1. The theorem statements and proofs are "not-rigorous". For example, theorem 1's main claim uses $\approx$ without formally setting in what sense is this an approximation. It seems to be that the equality is an almost-sure convergence as $n \to \infty$, and the approximation is for large-finite n ? Making this precise will be useful.

2. Similarly, Theorem 2's main sentence has a hypothesis that "if the privacy budge $(\epsilon, \delta)$ is neither too low nor too high". This is not made precise even in the supplementary materials where the proofs are provided.

---

### Note · Authors · 2022-06-14

**Comment:**

Thank you all for your feedback. I am honestly impressed by the clearness of the summaries that you wrote regarding this submission. I think most of the provided criticism indicate substantial weak points, and their revision will in particular improve overall rigorousness and strengthen the experiments.

We have decided to withdraw the paper since a thoroughly revised version will have drastic changes. If it was possible, we would be interested to resubmit this work to TMLR in the future.

**Withdrawal Confirmation:**

I have read and agree with the venue's withdrawal policy on behalf of myself and my co-authors.